# Iridophore apoptosis mediates socially-regulated developmental color pattern plasticity in an anemonefish

Laurie J. Mitchell[1]*, Saori Miura[1], Youjung Han[1], Jann Zwahlen[1], Camille A. Sautereau[1,2,3], Bruno Frédérich[4], Vincent Laudet[1,2,5]

1 Marine Eco-Evo-Devo Unit, Okinawa Institute of Science and Technology Graduate University, Okinawa, Japan, 2 CNRS IRL 2028 "Eco-Evo-Devo of Coral Reef Fish Life Cycle" (EARLY), Okinawa Institute of Science and Technology Graduate University, Onna-son, Okinawa, Japan, 3 Sorbonne Université, CNRS, Biologie Intégrative des Organismes Marins, BIOM, Observatoire océanologique de Banyuls-sur-Mer, Banyuls-sur-Mer, France, 4 Laboratory of Evolutionary Ecology, FOCUS, University of Liège, Liège, Belgium, 5 Marine Research Station, Institute of Cellular and Organismic Biology, Academia Sinica, Yilan, Taiwan

* laurie.mitchell@oist.jp

## Abstract

Understanding the developmental basis of phenotypic plasticity is key to unraveling the origins of biodiversity. In coral reef fishes, color pattern changes during ontogeny can serve adaptive functions, yet the mechanisms and ecological contexts shaping these transitions remain largely unknown. Here, we investigated color pattern development in the tomato anemonefish (*Amphiprion frenatus*), which exhibits transient posterior white barring during early juvenile stages. We demonstrated that the timing of bar loss is plastic and modulated by the social environment, where juveniles cohabiting with adult conspecifics exhibited bar loss ~24 days earlier than those isolated from adults. Through transcriptomic profiling, we identified gene expression changes implicating apoptosis- and autophagy-related pathways, as well as alterations in chromatophore development. Moreover, shifts in the expression of multiple thyroid hormone marker genes highlighted the potential neuroendocrinal integration of social cues that promoted bar loss. Ultrastructural analyses via transmission electron microscopy and in-situ assays indicated massive apoptosis of iridophores and associated dermal remodeling during the white-to-orange transition. The pharmacological inhibition of caspases delayed bar loss, confirming the functional role of programmed cell death. Behavioral trials revealed that adults responded differently to juveniles with/without the posterior bar, suggesting a role of transient barring in conflict avoidance during recruitment. Lastly, our evolutionary reconstruction of this plastic trait suggests that colony size is an important factor promoting this ontogenetic switch throughout anemonefishes. Our results provide compelling evidence for socially mediated plasticity in color pattern ontogeny with ecological and evolutionary implications for communication and species diversification in reef fishes.

**Data availability statement:** RNA sequencing data is available via NCBI Short Read Archive (SRA) under BioProject PRJNA1392056 (https://www.ncbi.nlm.nih.gov/bioproject/PRJNA1392056). Experimental data and original code have been deposited at Zenodo (https://doi.org/10.5281/zenodo.17973175).

**Funding:** This work was supported by the Japanese Society for the Promotion of Science (https://www.jsps.go.jp/) (P22768 to LJM; 22H02678 and 25K02328 to VL). The funders had no role in the study design, data collection and analysis, decision to publish, or preparation of the manuscript.

**Competing interests:** The authors have declared that no competing interests exist.

**Abbreviations:** BP, biological process; CTCF, corrected total cellular fluorescence; DEG, differentially expressed gene; GLMM, generalized linear mixed effects model; GO, gene ontology; JNK, Jun N-terminal kinase; TEM, transmission electron microscopy; TH, thyroid hormone; TUNEL, TdT-mediated dUTP nick end labeling.

## Introduction

Understanding the origins of biodiversity is a perpetual goal in biology with relevance beyond the species level [1,2]. A major driver of phenotype variation across individuals is developmental plasticity, or the ability of organisms to alter their developmental trajectories and/or phenotypic outcomes in response to environmental conditions [1,3,4]. Importantly, this plasticity can be adaptive, conferring fitness advantages by allowing organisms to modify traits in response to fluctuations of biotic [5,6] and abiotic factors [7,8].

Vertebrate pigmentation and color patterns are especially sensitive to ontogenetic conditions as they are constantly exposed to changing environments and directly subject to selection (i.e., predation, mate selection) [9]. Developmental change in color pattern is facilitated by morphological changes in chromatophore populations and their proportional abundance via homeostatic processes (i.e., regulating cell proliferation and death), and/or the dysregulation of factors contributing to pattern maintenance (e.g., cell-cell communication) (reviewed by Duarte and colleagues [10] and Kratochwil and Mallarino [11]). Among vertebrates, coral reef fishes have an extremely diverse array of color patterns both within and across species, making them ideal for studying variation of these phenotypic traits [12,13]. Their color patterns are important to survival and can be multipurpose (e.g., in camouflage and communication [14]), depending on background lighting and observer visual abilities [15,16]. Moreover, some reef fishes exhibit adaptive plasticity in color pattern during ontogeny in response to habitat [17–20], and social conditions [21,22]. Yet, despite their ecological relevance, the molecular and cellular mechanisms driving these color changes and their functional significance remain largely unclear.

Anemonefishes (Pomacentridae, Amphiprioninae) are a group of reef fishes that includes 29 species, all of which exhibit protandrous sex change and live in an obligatory symbiosis with various species of giant sea anemones [23]. Additionally, they are highly social fishes that form stable size-based social hierarchies within host anemones, where a strict rank order determines access to reproduction and shapes individual behavior [24–26]. Despite being evolutionarily young (~10 MY) [27], the adaptive radiation of anemonefishes has produced a variety of color patterns centered on a theme of zero to three white (vertical) bars and (horizontal) stripes against orange/red or black skin [28]. Underlying anemonefish skin colors are three broad types of chromatophores, including pigment-based cells (melanophores—black, xanthophores—orange) and iridophore cells containing bundles of reflective guanine crystal platelets that confer structural (white) coloration [29]. Bar pattern formation is coordinated by thyroid hormone (TH) following an anteroposterior order that marks the transition from the pelagic realm to settling onto the reef (studied in *Amphiprion ocellaris*) [28,29]. Color pattern diversity in anemonefishes has been evolutionarily linked to patterns in host (anemone) species usage [30], and there are several proposed functions in social signaling [31,32], species recognition [33,34], camouflage, and warning coloration [35]. One function that has gained attention is the ability of three barred anemonefish (*A. ocellaris*) to distinguish conspecifics from congeners based on differences in bar number (or total area) [34]. Indeed, the low (nonrandom)

overlap in bar pattern within sympatric anemonefish communities and in mixed species groups, highlights its role in promoting species coexistence and diversity [28,36].

The adaptive radiation of anemonefishes has led to repeated evolutionary losses of bars since the ancestral three barred state [28]. In many anemonefishes, bar loss occurs after initial formation during juvenile ontogeny in a consistent posterior-to-anterior sequence. While other species exhibit the direct (non-developmental) loss of bars, suggesting a common developmental program which became fixed in some species [28]. Previous studies identified the transcriptional markers of anemonefish chromatophores [29], and the importance of TH during bar formation [28,37], but the cellular processes and transcriptional changes mediating bar loss during ontogeny are unknown. Moreover, the adaptive value (if any) of transient barring remains elusive. Bar loss status in natural populations of anemonefish juveniles correlates with social cohabitation at host anemones, while supplemental bars tend to be exhibited by those which settle alone (Fig 1A). Thus, raising the idea that there may be developmental plasticity in the trajectory of bar loss with potential social importance. Both the HPA/corticoid and HPT/TH neuroendocrinal axes can respond to social conditions [38–40] and coordinate various homeostatic processes, that might control the loss of white skin.

Here, using an integrative eco-evo-devo approach, we showed a temporal plasticity in tomato anemonefish (*Amphiprion frenatus*) ontogenetic bar loss that depends on the environment and the programmed cell death (apoptosis) of dermal chromatophores. We found that juvenile anemonefish, dominated by adults in a host anemone, lost their posterior body bar earlier than compared to siblings without adults. Transcriptomic analyses identified changes in gene expression associated with iridophore and melanophore production, as well as programmed cell death and TH signaling. Transmission electron microscopy (TEM) revealed changes in iridophore abundance and cell morphology during the transition from white to orange skin consistent with mass apoptosis, which was confirmed using in-situ histological assays and an in-vivo pharmacological blockade of caspase protein activity. Finally, our evolutionary reconstruction of this plastic trait indicated that anemonefish living in smaller groups tend to harbor this plastic trait. Altogether, our results illustrate how white bar loss in juvenile anemonefish is shaped by socially regulated developmental plasticity of color patterns to suit natural contexts.

## Results

### Plasticity in the timing of bar loss responds to environmental conditions

In natural populations of three anemonefish species with one bar (*A. frenatus* and *A. melanopus*) or two bars (*A. chrysopterus*) as adults, we observed that juveniles frequently exhibited supplemental bars or a reduced (adult) pattern according to their social isolation or cohabitation with conspecifics at host anemones, respectively (Fig 1A). To address whether developmental plasticity underlies this variation in color pattern and identify the external factors (habitat quality, social cues) driving this change, we experimentally assessed the rate of bar loss in one of these species, *A. frenatus*, raised in different environmental conditions. We intermittently monitored the degree of bar loss in juvenile anemonefish sampled at 38 and 62 days-post-hatching (dph) in two replicates (n = 8 per age group) that were kept in four conditions: (*1*) with a live sea anemone occupied by two adults, (*2*) with a live sea anemone without adults, (*3*) with fake/plastic anemones without adults, and (*4*) with no habitat in an empty tank (Fig 1B). Treatments 1 and 2 recreated the natural settlement conditions experienced by juveniles returning to the reef, where they settle into either an inhabited or uninhabited host. This isolated the effect of top-down social pressure on bar loss by adult presence/absence in natural habitat. In addition, the inclusion of treatments 3 and 4, isolated the importance of a live host (symbiosis) from the availability of structural habitat (shelter). Juveniles entered both alive and plastic anemones. They were also tolerated by adults, but mainly kept at the periphery of anemones by chasing and intimidation. After 20–90 days, depending on the environmental condition, bar loss was initiated by the tapering and thinning of the ventral-end of the bar, followed by the dorsally faded retreat of the white skin and flanking (black) edges (Fig 1C).

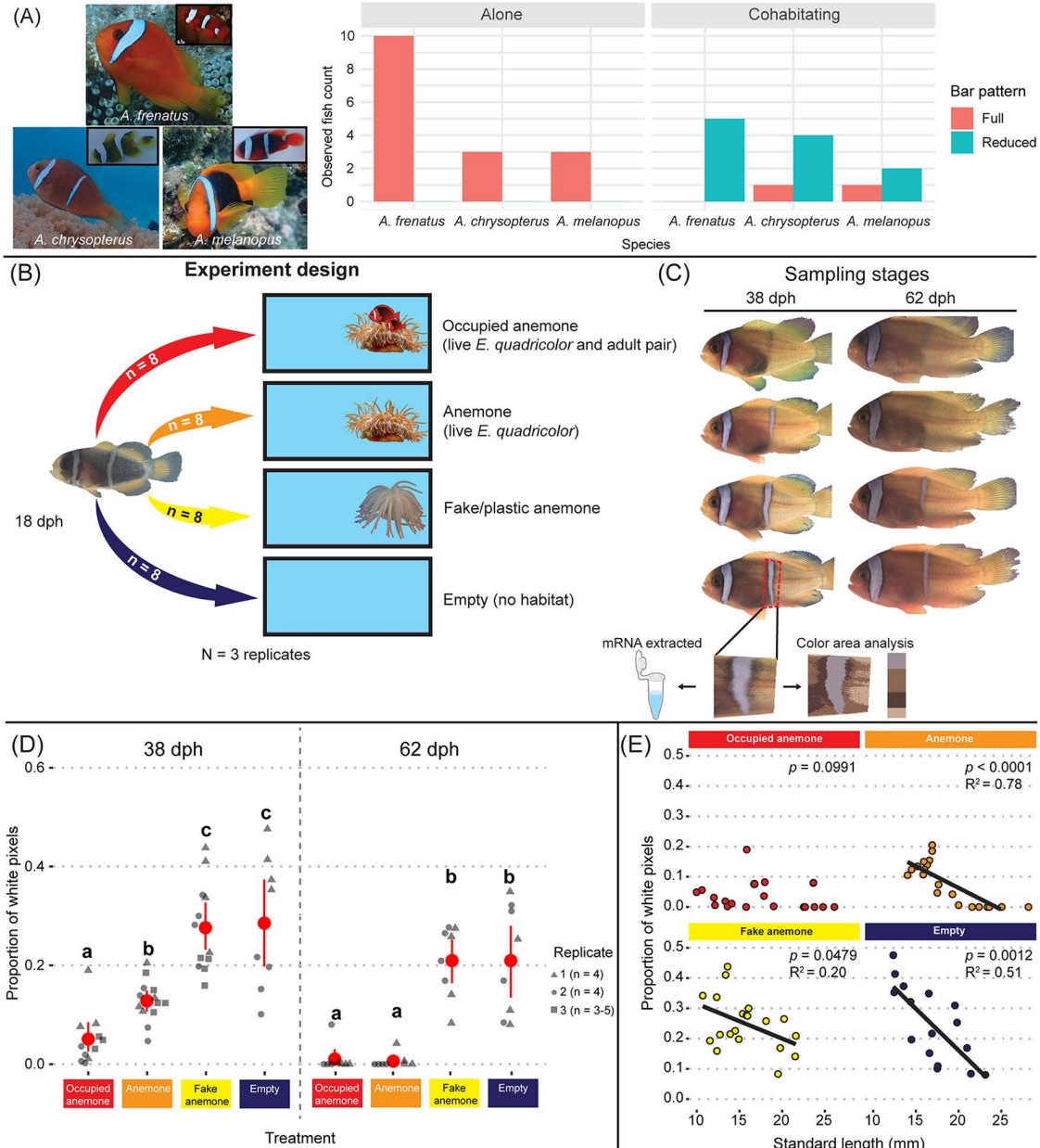

**Fig 1. Environment influenced the timing and extent of white bar loss in anemonefish juveniles. (A)** Field observations (juvenile counts) of three species with bar loss during ontogeny and their state of bar pattern (full or reduced) according to sociality (alone or cohabitating) at host anemones. **(B)** *Amphiprion frenatus* (*n* = 32 per replicate) were transferred at 18-days-post-hatch (dph) into four aquaria with distinct environments. **(C)** Shown are representative individuals from each treatment at 38 dph and 62 dph. Remaining white skin area was calculated per juvenile as the proportion of white pixels out of the total pixel count in a fixed region containing the body bar. Skin mRNA samples were extracted using both sides of juveniles from Replicate 1. **(D)** White skin area calculated from images taken of the left side body bar region. Red points and intervals denote the mean (±0.95 CI). Letters denote statistically significant (ordbeta reg, *p* < 0.05) grouping between treatments. The third replicate was conducted till 38 dph and excluded the Empty treatment. **(E)** Individual proportions of white skin area in each treatment combined by age and plotted against standard length (mm). Reported *p*-values and multiple R-squared values were from linear regression analyses. The data underlying this Figure can be found in https://doi.org/10.5281/zenodo.17973175.

Ordered beta (ordbeta) regression analysis on the relative area of the white body bar yielded significant variation ($p<0.05$) among multiple treatments by sampled age group (Fig 1D, S1, and S2 Tables). A Tukey HSD multiple comparisons test indicated that at day 20 (38 dph) juvenile tomato anemonefish which cohabited with adults had a significantly lower mean proportion of white skin (mean $\pm$ SEM = $0.051 \pm 0.016$; Fig 1D) than juveniles kept with just an anemone ($0.13 \pm 0.012$) (OR = 0.22, SE = 0.06, $z$-ratio = −5.24, $p_{adj}<0.0001$), fake anemone ($0.3 \pm 0.026$) (OR = 0.13, SE = 0.03, $z$-ratio = −8.11, $p_{adj}<0.0001$), and empty tank ($0.29 \pm 0.048$) (OR = 0.11, SE = 0.03, $z$-ratio = −7.88, $p_{adj}<0.0001$). These differences were clearly visible amongst treatments (Fig 1C). Juveniles kept with only an anemone also exhibited a significantly lower proportion of white skin than the Fake anemone treatment (OR = 0.57, SE = 0.12, $z$-ratio = −2.74, $p_{adj}=0.032$) and Empty treatment (OR = 0.51, SE = 0.11, $z$-ratio = −3.07, $p_{adj}=0.012$). Juveniles at 38 dph in the Fake anemone and Empty treatments had comparable white skin cover to the pretreatment ($0.37 \pm 0.028$). Due to high mortality of juveniles in the third replicate ($n=3$−5 survivors), it was limited to 20 days and had to drop one treatment (Empty). By about 62 dph, the juveniles kept in the Anemone treatment had a reduced area of white skin in the body bar region on-par with that of most individuals in the Occupied Anemone treatment, where little-to-no white skin remained (combined range = 0–0.08; Fig 1C, 1D). There remained a significantly lower area of white skin in both live anemone containing treatments than in the Fake Anemone ($0.21 \pm 0.026$) and Empty ($0.21 \pm 0.040$) treatments (ORs = 0.02–0.08, $z$-ratios = −3.96–−2.59, all $p_{adj}<0.05$; Fig 1D). Approximately 5 to 6 months passed until no white skin was visible in the body bar region of remaining (non-experimental) siblings that were maintained in equivalent conditions to the Empty treatment.

Overall, and as expected, the proportion of white skin also significantly decreased with juvenile body size (standard length/SL) (ordbeta, $n=76$, $\beta=-0.50$, SE = 0.14, $z$-value = −3.66, $p=0.00025$). Regular linear regression showed a negative relationship between white skin area and body size in all treatments, except for the Occupied Anemone (Fig 1E).

Because growth in anemonefish can adjust in response to both the anemone symbiosis and social pressure [24,41], we compared juvenile body sizes (in standard length/SL) among the environmental treatments (S1A, S1B Fig). Although statistically nonsignificant, the mean ($\pm$SD) SL at 38 dph in replicate 1 was larger in the Anemone treatment ($17.70 \pm 1.07$ mm) and Empty treatment ($16.43 \pm 1.64$ mm) than in the Occupied anemone treatment ($14.20 \pm 2.57$ mm) and Fake anemone treatment ($13.80 \pm 2.72$ mm). A shift in body size was observed at 62 dph, where mean SL was significantly larger in both live anemone containing treatments (22.53–28.18 mm) than the Fake and Empty treatments (14.38–21.59 mm) (ANOVA, $F_{3,12}=11.55$, $p=0.013$). In replicates 2 and 3, body sizes at 38 dph were significantly larger (ANOVA, all $p<0.01$) in treatments containing live anemones (15.00–17.96 mm) than in the Fake anemone and Empty treatments (11.54–14.49 mm).

Taken together, these data indicate that the rate of white bar loss was mainly linked to the status of a live host, and to a lesser extent to the presence of adults, while only marginally to juvenile body size. Importantly, our results suggest that the juvenile fish, in response to perceiving their socially enriched environment, accelerated their transition from the two-bar to the one-bar phenotype.

## Skin transcriptomic changes are associated with bar loss and body size

To identify the skin-level processes underlying this color pattern change, we performed transcriptomic analyses using skin and attached muscle tissue from the body bar region of juvenile tomato anemonefish ($N=36$ from Replicate 1) (Fig 2A, 2B). We observed that the environmental conditions which contained live anemones (Occupied anemone and Anemone treatments) produced a less heterogenous transcriptomic response in the skin of 38 dph juveniles (Fig 2B).

We performed a PCA on all expressed genes in 38 dph samples ($n=16$; Fig 2A) to visualize the overall sample distribution and transcriptomic effects by environmental treatment. The first two axes captured 40.39% of total mRNA expression variation, where PC1 (25.30% of variation) was mainly associated with fish body size, as evident by the broad separation of samples according to juveniles that were larger (namely Anemone and Empty treatments; negative scores along PC1; mean SL $\pm$ SD = $17.50 \pm 0.82$ mm, $n=7$) and smaller (namely Fake anemone and Occupied anemone; positive scores along PC1; $14 \pm 2.30$ mm, $n=9$). To account for size differences that potentially influenced juvenile behavior, we calculated

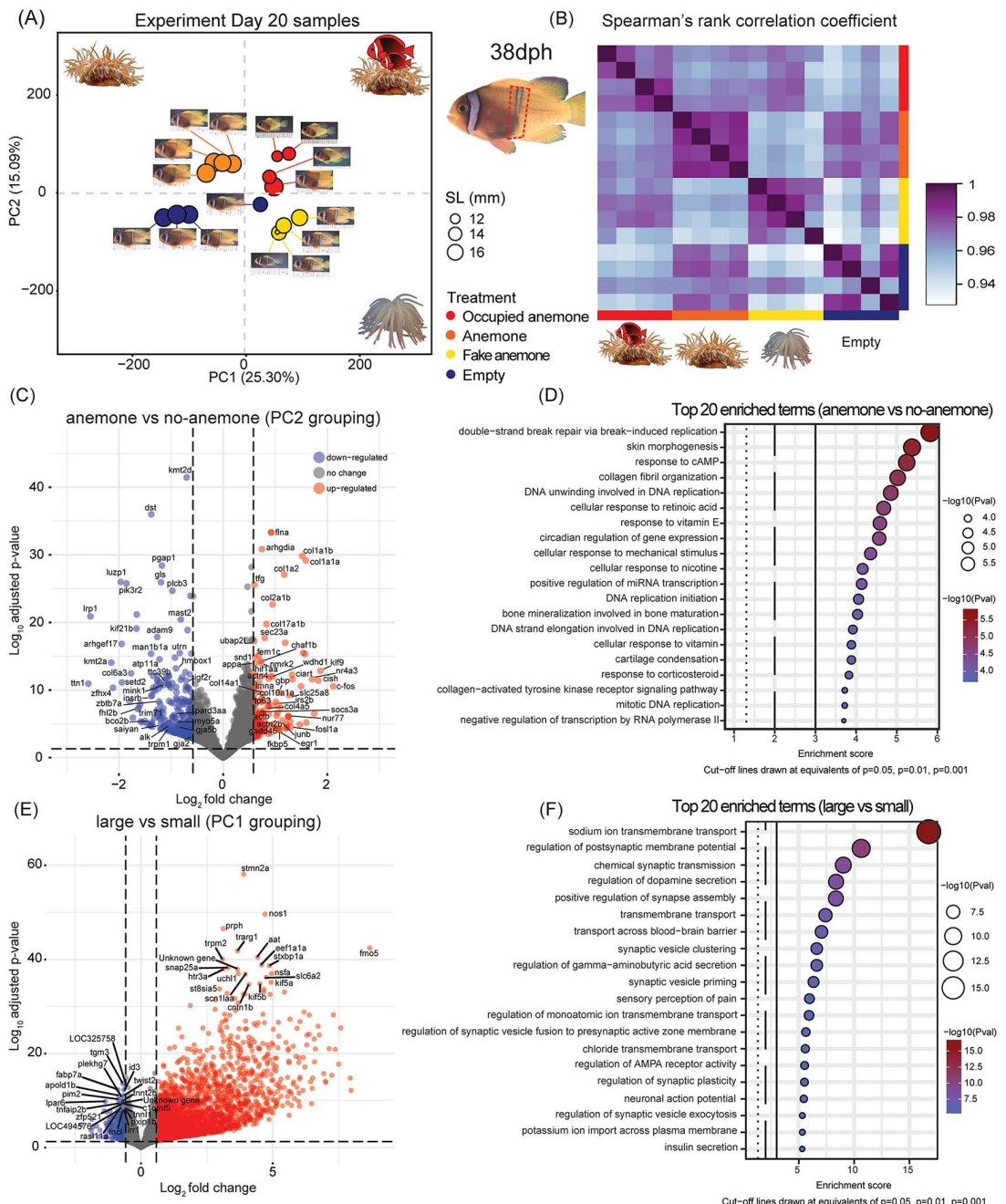

**Fig 2. Live anemone presence and body size shape transcriptomic signatures in tomato anemonefish skin. (A)** PCA showing the distribution of Day 20 (38 dph) samples along PC1 and PC2 according to total gene expression with point size in proportion to fish standard length (mm). **(B)** Correlation matrix depicting pairwise Spearman's rank correlation coefficients, calculated using transcriptomic data. **(C)** Volcano plot depicting the top differentially expressed genes (log fold change ≥1.50, $p$-value < 0.05) between the treatments, which included a live anemone ("Occupied anemone" and "Anemone") and those without ("Fake anemone" and "Empty"). **(D)** Top 20 gene ontology (GO) terms for Biological Processes (BP) returned from a GO-enrichment analysis based on the presence of a live sea anemone. **(E)** Volcano plot showing top differentially expressed genes between juveniles with larger and smaller mean body sizes (samples grouped by PC1). **(F)** Top 20 GO terms enriched in larger-bodied juveniles. The data underlying this Figure can be found in https://doi.org/10.5281/zenodo.17973175.

the size ratios in each treatment of smaller and larger juveniles (i.e., those adjacent in rank). The mean (±SE) size ratios were higher in treatments with larger SLs (Anemone = 0.96 ± 0.009; Empty = 0.96 ± 0.019) than with smaller SLs (Occupied Anemone = 0.90 ± 0.024; Fake Anemone = 0.92 ± 0.027), meaning that juveniles were not only larger on average in the former treatments but also closer in size implying weaker top-down social pressure. PC2 (15.09% of variation) was mainly associated with the degree of body bar loss and whether the treatment contained a live anemone (positive scores along PC2; mean proportion of white skin ± SD = 0.048 ± 0.053, $n = 8$) or no live anemone (negative scores along PC2; 0.25 ± 0.081, $n = 8$).

A second PCA, performed on expressed genes with 62 dph samples (total variation = 30.47%, $N = 16$; S2 Fig), produced similar clusters than observed for 38 dph samples but with more overlaps. The mean size ratios between juveniles at 62 dph were similar across all treatments (~0.90).

An analysis of the common differentially expressed genes (DEGs) in skin samples grouped according to the presence of a live anemone (i.e., PC2 clustered samples) yielded a total of 144 upregulated genes and 294 downregulated genes (Fig 2C). Half of the top 10 upregulated genes were identified as belonging to the collagen family (*col1a1a/b*, *col1a2*, *col2a1b*, *col17a1b*) with roles in fibrillar collagen production. Additional collagen production genes upregulated in the anemone containing treatments included *col4a5*, *col10a1a*, and *col14a1*. Among the top 10 upregulated genes were also those with various functions, such as actin filament and cytoskeleton (*flna*), endoplasmic reticulum to Golgi transport (*tfg*, *sec23a*), and stress response signaling (*ubap2l*). Other upregulated genes were identified with mixed roles in apoptosis and proliferation (*c-fos*, *junb*, *fosl1a*, *tp63/p51*, *gadd45*, *nur77*, *egr1*), while three of the top 10 downregulated genes had various roles in cell survival, proliferation, and growth (*kmt2d*, *pik3r2*, *dst*). Chromatophore-related genes were also detected among the downregulated genes (Fig 2C), including those with roles in iridophore formation (*fhl2b*, *alk*, *Saiyan*) [29,42,43], ketocarotenoid (red pigment) production (*tt3c9b*) [44,45], and melanogenesis (*myo5a*) [46]. A gene ontology (GO) enrichment analysis of biological process (BP) terms analyzed the functional significance of the DEGs. Most of the upregulated collagen genes were represented in 12 of the top 20 enriched BP terms (Fig 2D) such as "skin morphogenesis" (GO:0043589, $n = 4$ significant genes), "collagen fibril organization" (GO:0030199, $n = 7$), "cellular response to mechanical stimulus" (GO:0071260, $n = 6$), "response to corticosteroid" (GO:0031960, $n = 10$), and "collagen-activated tyrosine kinase receptor signaling pathway" (GO:0038063, $n = 3$).

An analysis of the DEGs in samples grouped by body size (i.e., PC1 clustered samples) yielded 2,571 upregulated genes and 284 downregulated genes in larger-bodied juveniles (Fig 2D). Among the top 10 upregulated genes were those with roles in neuronal growth and/or axonogenesis (*stmn2a*, *prph*), neurotransmission (*stxbp1a*, *snap25a*, *htr3a, aat*), transmembrane action potential (*trpm2*), carbohydrate metabolism (*trarg1*, *sta8sia5*, *fmo5*), wound healing and angiogenesis (*nos1*). The top 20 enriched BP terms from a GO term enrichment analysis contained 12 with overlapping functions in synapse activity and/or assembly ($N = 239$ significant genes), six transmembrane transport terms ($N = 269$), "insulin secretion" (GO:0030073, $n = 50$), and "sensory perception of pain" (GO:0019233, $n = 32$) (Fig 2E).

These results demonstrate that the presence of a live anemone as well as body size independently structured the transcriptomic landscape in juvenile tomato anemonefish skin. These factors distinctly influenced gene programs related to pigment cell regulation, particularly but not limited to iridophores, tissue remodeling, and apoptosis during bar loss.

## Broad downregulation of chromatophore-related genes during bar loss

A more focused analysis on a subset of genes ($N = 220$) with relevance to chromatophore development and color pattern formation in teleost fishes and other vertebrates highlighted numerous differentially expressed genes in addition to some already mentioned (Fig 3A).

Five marker genes for iridophores (*saiyan*, *fhl2b*, *alk*, *ltk*, *apod*) were significantly downregulated in the fading body bar region of 38 dph juvenile tomato anemonefish from both live-anemone treatments compared to Empty and/or Fake anemone (Fig 3A). These four genes remained downregulated at 62 dph, with a significantly lower expression of *fhl2b*

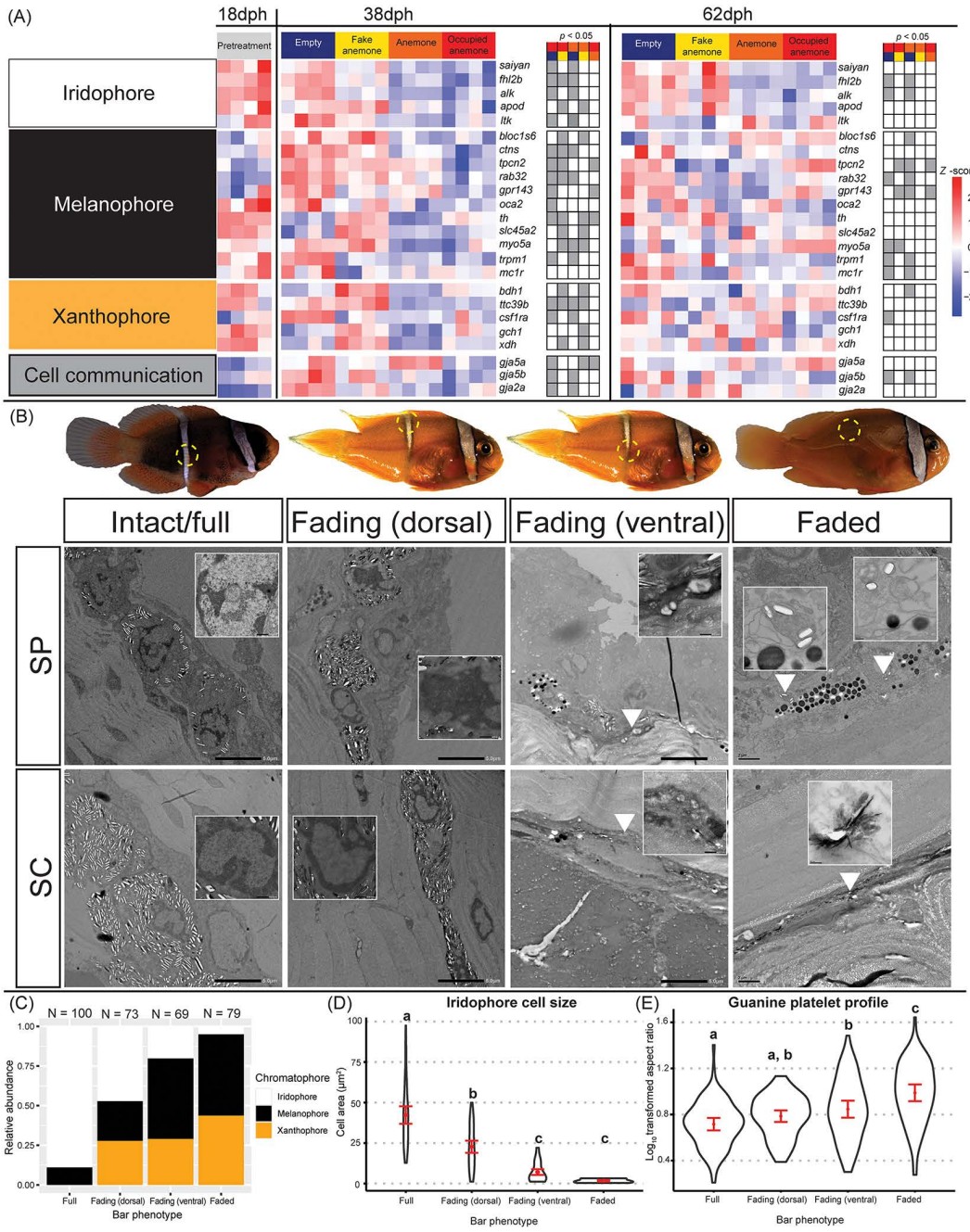

**Fig 3. The cellular underpinnings of white bar loss reveal iridophore reduction and coordinated pigment cell remodeling. (A)** Heatmaps depicting the normalized gene count expression patterns (z-scores) related to chromatophore formation, thyroid hormone signaling, and other cellular functions, according to sampling age (dph) and (pre-)experimental treatment. Gray markers indicate pairwise statistical significance (Wald Test, p-value < 0.05) between paired (color-coded) treatment comparisons. **(B)** TEM images of iridophores (x1.5k mag.) in cross-sectioned skin taken from the body bar region of *A. frenatus* during three broad stages of bar loss (intact/full bar, $n = 3$ fish; fading, $n = 2$; faded, $n = 1$). Iridophores were observed in both the stratum spongiosum, "SP" and inner stratum compactum, "SC" dermal layers. Two areas were sectioned in the fading white skin to compare the chromatophore populations dorsally in the "solid" white skin and ventrally at the receding edge of the white skin. Inset images depict closeups (x8–10k mag.) of iridophore nuclei. White arrowheads indicate small iridophores. **(C)** Overall proportional abundance of each chromatophore type calculated out of a total cell count (N) across three sections per bar loss stage/phenotype (except fading dorsal, $n = 2$). **(D)** Violin plots showing the mean (±0.95 CIs) iridophore cell area (µm²) and **(E)** log-transformed aspect ratios (length/width) of guanine platelets per bar loss stage. Letters denote groups of statistical significance (ANOVA, p-value < 0.05). The data underlying this Figure can be found in https://doi.org/10.5281/zenodo.17973175.

and *alk* in both anemone-containing treatments than the Empty treatment. The presence of a host anemone was also associated with the downregulation of 11 melanophore-related genes, including the *oca2* gene that regulates melanin production in multiple vertebrates [47–50]. Two marker genes for xanthophores (*gch1*, *xdh*) [51], and ketocarotenoid production (*bdh1*, *ttc39b*) [45,44] were also upregulated at 38 dph in the Fake Anemone treatment than compared to both anemone-containing treatments (Fig 3A). This data suggests that environmental and social contexts, including the presence of a host anemone, triggered a coordinated transcriptional response across all three chromatophore cell types, not just iridophores, ultimately driving the observed differences in coloration. This broad regulatory shift highlights the integration of ecological cues into the genetic control of color pattern development. Of note, in the samples from both anemone-containing treatments, we detected the downregulation of two gap junction proteins, *gja5b* and *gja2a* (Fig 3A), the former of which has functional importance in stripe pattern formation in zebrafish (*Danio rerio*) [52,53] and anemonefish (M. Klann, personal communication).

To relate gene expression changes to the state of chromatophores in the skin, we analyzed TEM images of cross-sectioned white skin in the body bar region. The fully intact bar at 18 dph contained dense aggregations of large iridophores that were unambiguously identified by their crystal platelets (Fig 3B–3D). This appearance was reminiscent of previously examined sections in the head bar [54], where the predominant chromatophore type in the dermis was iridophores (~90%), interspersed with small melanophores that were observed as deep as the hypodermis. During bar loss, there was a noticeable shift away from iridophores towards melanophores and the introduction of xanthophores (Fig 3B, 3C). ANOVA conducted on the normalized total area of iridophores measured in TEM images revealed a significant difference in iridophore cover among three varying states of bar loss ($F(3, 15) = 16.07$, $p < 0.0001$). A Tukey HSD multiple comparisons test indicated a significant decrease ($p = 0.0028$, 95% CI [−47.92, −15.26]) in iridophore cover between the full bar (mean ± SE = 35.71 ± 5.55, $n = 3$ sections) and the ventrally sectioned fading bar (4.12 ± 3.47, $n = 3$). In the faded white skin only three small iridophores were found in two out of the three sections. ANOVA also indicated significant differences in individual iridophore cell area during bar loss ($F(3, 137) = 44.21$, $p < 0.0001$; Fig 3D). A paired comparisons test indicated that iridophores were significantly smaller in the fading bar (dorsal = 22.75 ± 1.60 $\mu m^2$, $n = 35$; ventral = 7.09 ± 0.77 $\mu m^2$, $n = 69$) and faded bar (1.78 ± 0.84 $\mu m^2$, $n = 3$) than in the full bar (42.31 ± 2.06 $\mu m^2$, $n = 89$) (Tukey HSD, all $p < 0.0001$, Fig 3D). The profile of guanine platelets inside iridophores also significantly varied among the bar loss phenotypes ($F(3, 374) = 22.37$, $p < 0.0001$; Fig 3E). The mean aspect ratios of platelets were significantly higher in both the ventral area of the fading bar (0.85 ± 0.03, $n = 100$; $p_{adj} = 0.0004$) and faded bar (0.99 ± 0.03, $n = 78$, $p_{adj} < 0.0001$) than in the full white bar (0.72 ± 0.02, $n = 100$), meaning that guanine platelets had become narrower during bar loss.

In case the loss of white skin was due to the dispersal of iridophores outside the body bar region, we also analyzed TEM images taken in adjacent orange skin but found no iridophores (S3 Fig). Our data supports a model in which bar loss is driven by a coordinated reduction in iridophore number, size, and internal structure, rather than dispersal or obscurement – highlighting a precise and dynamic cellular remodeling process in response to environmental cues. In addition, our transcriptomic data suggest this is a tightly regulated process involving all chromatophore types, underpinned by cellular reorganization and hormonal regulation.

## Apoptosis-mediated cell death of chromatophores is required for bar loss

In addition to a decreased iridophore abundance and size during bar loss, we observed that their nuclei often had enlarged dark patches of condensed chromatin either at the periphery of the nuclear envelope or encompassing the entire nucleus (Fig 3B, insets). Iridophores also exhibited compressed cell shapes and/or fragmented nuclei, and in the faded skin, only loose guanine platelets were found in small membranes. This raised the hypothesis that iridophore cells were perhaps dying and being replaced by pigment cells (xanthophores and/or melanophores).

Our hypothesis was supported by an analysis of the normalized counts of a subset of genes ($N = 111$) with known proapoptotic and/or autophagy function(s). Indeed, we detected 28 upregulated orthologs in the skin samples of 38 dph

tomato anemonefish in both live-anemone treatments and especially in the Occupied anemone treatment (Fig 4A). Most of these genes were captured within four positively enriched GO terms related to apoptosis (GO:0043065, GO:0043523, GO:0051402, GO:0043281). Other positively enriched GO terms relevant to cell death were found in the live-anemone treatment samples, including: "regulation of reactive oxygen species biosynthetic process" (GO:1903426), "response to oxidative stress" (GO:0006979), and "negative regulation of cell-substrate adhesion" (GO:0010812). The Caspase-3 gene (*casp3*) that encodes a major effector protein at the execution stage of apoptosis was significantly upregulated in the Occupied Anemone treatment samples compared to all other treatments. Among other upregulated genes, we identified three orthologs of known transcription factors of the (pro-apoptotic) tumor suppressor gene, *p53* (*tp63*, *aspp1*, *sik1*), phosphorylators/activators of *p53* (*noxa*, *jnk*), and other genes which are activated by interaction with *p53* (*ntrk1*, *jmjd2b*). Although statistically nonsignificant, a general inversion in the pattern of pro-apoptotic gene expression was observed at 62 dph, that coincided with the completion of bar loss in the anemone-containing treatments and its onset in the anemone-deprived treatments.

We also detected in the skin of 38 dph juveniles from the Occupied Anemone treatment, the upregulation of five orthologs of pro-autophagy genes, including *gabarapl2*, *map1lc3a*, *ulk1a*, *ulk2*, and *uvrag* (Fig 4A). Thus, apoptosis and autophagy may possibly act in parallel. All these observations suggest that apoptosis-related pathways may be temporally and spatially regulated in a context-dependent manner, aligning with the timing of the observed white bar loss.

To test for apoptosis in the fading white skin, we performed a TdT-mediated dUTP nick end labeling (TUNEL) assay in three cross-sectioned dorsoventral skin regions. This assay fluorescently labeled nuclei containing DNA double-strand breaks, a characteristic of late-stage apoptosis. We detected TUNEL+ (fluorescent) cells in the dermal layers of three cross-sectioned skin regions (Fig 4B). Only in the positive control that was incubated with Nuclease S7 was fluorescence detected in the hypodermis. Peak corrected total cellular fluorescence (CTCF) was detected in the mid-region sections located at the edge of the receding white bar (mean $\pm$ SE $= 8.00e^{+6} \pm 6.18e^{+5}$), while it was dimmest in dorsal sections ($9.27e^{+5} \pm 1.52e^{+5}$). TUNEL+ cells were still detected in late-stage faded white skin (S4 Fig), but with an overall lower CTCF intensity that peaked in the dorsal section. This diminished signal further demonstrated that higher levels of apoptotic activity were linked to the event of bar loss and dying of iridophores.

To further validate apoptotic activity, we immunolabelled cleaved (activated) caspase-3 protein with DAB staining in the dermis of cross-sectioned skin ($n = 3$ sections per region, $N = 2$ fish). Positive (DAB+) stained regions were clearly observed in the mid-region, and to a lesser degree in the ventral region (red arrowheads in Fig 4C). No DAB+ cells were observed in dorsal sections, nor in sections made in orange skin (S5 Fig). While in these classical histology sections, it was not possible to identify iridophores based on the presence of guanine platelets, we could exclude DAB+ cells being melanophores or xanthophores due to the absence of dark melanocytes and yellow/orange xanthosomes, respectively.

These results provide direct evidence of localized apoptosis in the fading white skin, particularly at the receding edge of the bar, supporting a role for programmed cell death in the active remodeling of chromatophore populations during bar loss.

## Pharmacological impairment of bar loss by a broad caspase inhibitor

To directly demonstrate the contribution of apoptosis to bar loss, we compared the level of white skin loss among groups of juveniles that were either fed with food soaked in a broad caspase inhibitor (Z-VAD-FMK) or DMSO (control). After 26 days of drug exposure, we compared the area of white skin (pixels) loss using landmark-aligned images of the body bar region. Wilcoxon signed rank tests indicated that the median area of white skin significantly decreased between Days 1 and 26 within both treatment groups and the control (all $p < 0.05$; Fig 4D).

A dose-dependent response was evident in the degree of white skin loss, with the highest decrease in the control (mean $\pm$ SE $= 0.38 \pm 0.05$, $n = 16$), that was followed by the low dosage ($0.27 \pm 0.04$, $n = 8$) and then high dosage ($0.19 \pm 0.03$, $n = 16$) (Fig 4E). Ordbeta regression analysis and a Tukey HSD multiple comparisons test indicated that the

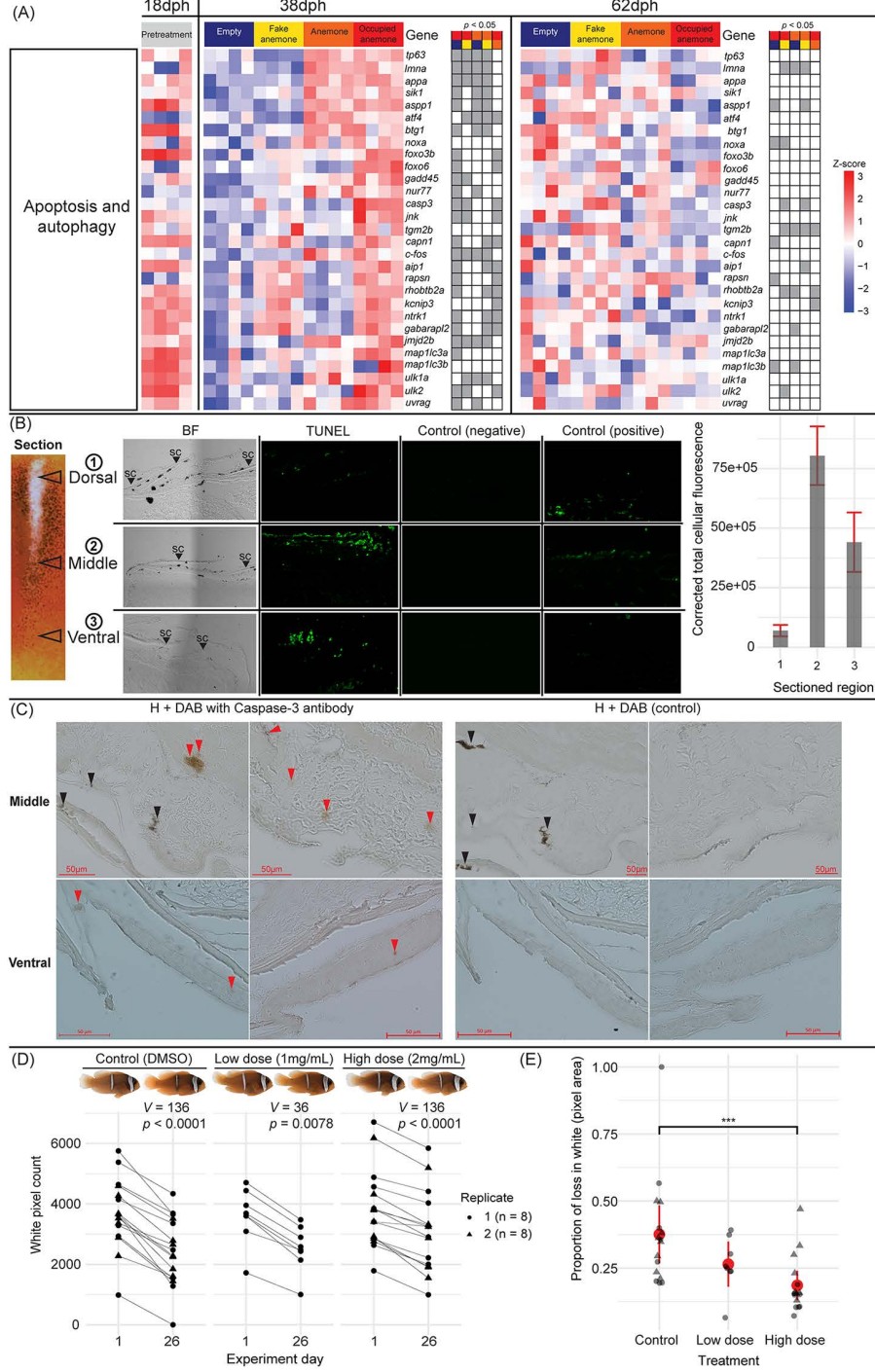

**Fig 4. Apoptosis is activated during white bar loss and contributes to its progression. (A)** Normalized and centered gene count expression patterns (z-scores) according to sampled age (dph) and environmental treatment for genes with known roles in apoptosis and/or autophagy. Gray markers indicate pairwise statistical significance (p < 0.05) between (color-coded) treatment comparisons. **(B)** TUNEL assay micrographs in three sectioned regions of the fading body bar in *A. frenatus*, including brightfield "BF", TUNEL reaction mix-treated, negative control, and positive (Nuclease S7) control sections. The side plot depicts the mean (±0.95 CI) area corrected total cellular fluorescence (CTCF) calculated for individual cells in three sections per the ventral (n = 66), middle (n = 77), and dorsal (n = 74) regions. **(C)** Brightfield micrographs from immunostaining using cleaved/activated Caspase-3 antibody with DAB in the cross-sectioned mid and ventral regions of *A. frenatus* skin, and in corresponding control sections, which lacked antibody. Sections were counterstained with hematoxylin "H". Arrowheads indicate DAB+ regions (red) and melanophores (black). **(D)** Pre- (Day 1) and post- (Day 26)

treatment exposure to a broad caspase inhibitor (Z-VAD-FMK) or DMSO (control), comparing the relative area of the white body bar (white pixel count/ total pixel count) in fish ($n=8$ per replicate). Statistics are given from the Wilcoxon signed rank tests. **(E)** Mean (±0.95 CI) proportional change in white body bar (pixel) area [(initial pixel count – final pixel count)/ initial pixel count] by pharmacological treatment/control. Only the high dose and control were included in the second replicate. Note that independently raised fish were used in the DGE analysis, TUNEL assay, and immunostaining. "***"=$p<0.001$. The data underlying this Figure can be found in https://doi.org/10.5281/zenodo.17973175.

mean proportion of loss in white area was significantly lower in the high dosage group than in the control (OR =0.46, SE =0.09, z-ratio =−4.07, $p_{adj}$=0.0001; Fig 4E). One outlier individual from the control (ID C7) had completely lost its body bar by Day 26 and gave an extreme value of 1.00. However, dropping this individual from our analysis did not affect the statistical significance of the results (S3 Table).

These data show that the inhibition of caspase-3 activity slowed down bar loss, clearly demonstrating that apoptosis is a key driver of this socially and environmentally controlled ontogenetic color pattern change.

## A systemic thyroid hormone upregulation by social cues

The idea that juvenile fish responded to enriched social conditions by accelerating bar loss, implies there should be a systemic global signal linking the brain to the detection (via sensory organs) and integration of social cues to drive changes in the skin. To identify this signal, we combined our (local) skin DGE analysis with a global transcriptomic analysis of entire juveniles cohabiting with or without an adult pair. We focus our analysis on two signaling pathways that could be affected by social cues: the stress HPI/corticoid axis, as it is well known that social interactions between adults and juvenile fish involve frequent aggressive interactions [26], and the HPT/THs axis, as these are linked to white bar formation [20].

In the skin, we identified multiple differentially expressed genes at 38 dph with known roles in the TH signaling pathway, including *dio3a*, *trβ*, *klf9*, and *med1* (Fig 5A). The iodothyronine deiodinase 3 gene (*dio3a*) that converts active TH to an inactive state was significantly downregulated in the Occupied Anemone treatment when compared to the Anemone treatment and Empty treatment. The TH receptor beta (*trβ*) gene was also significantly downregulated in samples from the Occupied Anemone treatment than in both non-anemone treatments. Conversely, the Krüppel-like Factor 9 (*klf9*) gene, a classical target gene of the TH signaling pathway, was significantly upregulated in both live-anemone treatments than the Empty treatment. Similarly, the mediator subunit 1 (*med1*) gene, a coactivator of TH receptors, was significantly upregulated in both live-anemone treatments.

At the whole-body level, we found that the distribution of samples was slightly more heterogeneous, but the presence/ absence of adults largely explained variation in total gene expression (PC1 variation =21.18%, $n=5$; S6 Fig). Our global transcriptomic result did not indicate any coherent change of expression in the main genes of the HPI/corticoid axis that could be indicative of a global difference in stress level. In contrast, at the HPT/TH level, there was a clear separation of samples by treatment (Dim1 variation =25.50%; Fig 5B). Juveniles which cohabited with adults were found to up-regulate three key genes of the pathway, namely thyroglobulin (*tg*), one of the thyroid hormone receptors (*trab*), and, albeit statistically nonsignificant thyroperoxydase (*tpo*) (Fig 5B). Additionally, the paired box 2 (*pax2*) gene important in the development of thyroid follicles [55], was upregulated.

Altogether, these data suggest that juveniles which cohabited with adults had locally more active TH signaling, exemplified by decreased active TH degradation and high target/coactivator gene expression. While decreased TH receptor (*trβ*) expression levels in skin might have been regulatory feedback in response to overly high (global) TH production.

## Juveniles with a second white bar were attacked less by adults

Given the strong influence of the adult presence in accelerating bar loss, we next tested whether the supplementary white bar in juveniles plays a role in social interactions within tomato anemonefish colonies. To determine whether adult anemonefish recognized juveniles by differences in their white barring, we conducted behavioral experiments that compared

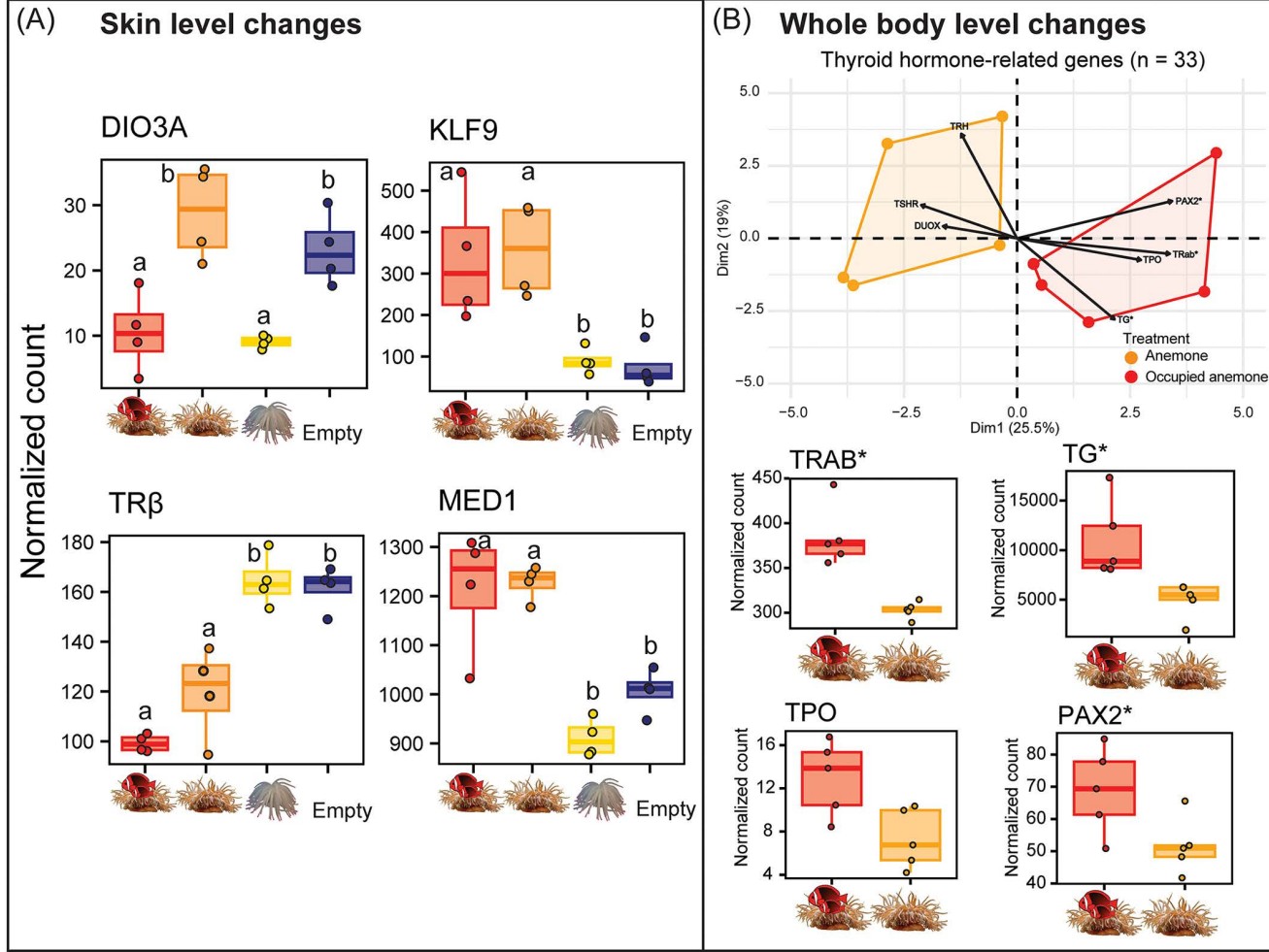

**Fig 5. Social cohabitation with adults triggered changes in thyroid hormone-related gene expression. (A)** Changes in TH-related gene expression detected in white/fading bar skin of (38 dph) juvenile tomato anemonefish ($n=4$) in each environmental condition. Letters denote groups of statistical significance. **(B)** Above; biplot showing the distribution of whole-body mRNA samples of 44 dph juveniles ($n=5$) kept with/without adults according to TH-related gene expression. Below are normalized expression levels of upregulated genes highly correlated with adult cohabitation. Box plots show the median, 25th and 75th percentiles, and range (whiskers). "*" = log fold change ≥1.0, and $p<0.05$. The data underlying this Figure can be found in https://doi.org/10.5281/zenodo.17973175.

adult aggression towards size-matched juveniles with a different number of bars (Fig 6A). During trials, the juveniles were presented to adults in-parallel from behind clear containers, and the number of attempted attacks (charges and bites) by adults was analyzed. Because males and females made a comparable number of attacks towards juveniles (S7 Fig), we combined data within pairs for analyses.

First, we compared the number of attacks by adults towards size-matched, one bar and two barred tomato anemonefish juveniles ($N=38$ fish, ΔSL mean±SD = 1.66±0.97 mm). The mean (±SE) number of attacks received by one bar juveniles (8.52±1.29) per trial was about 2.5 times higher than by two barred juveniles (3.17±0.57). The presence of a second bar in juveniles was associated with a significantly lower proportion of attacks from adults (binomial GLMM, $n=48$, estimate = −2.15, SE = 0.26, $z=−8.31$, $p<0.0001$; Fig 6B, S4 Table). To verify that the difference in adult responses was towards the white barring and not uncontrolled variation in surrounding (orange and black) pigmentation, we also compared the number of attacks

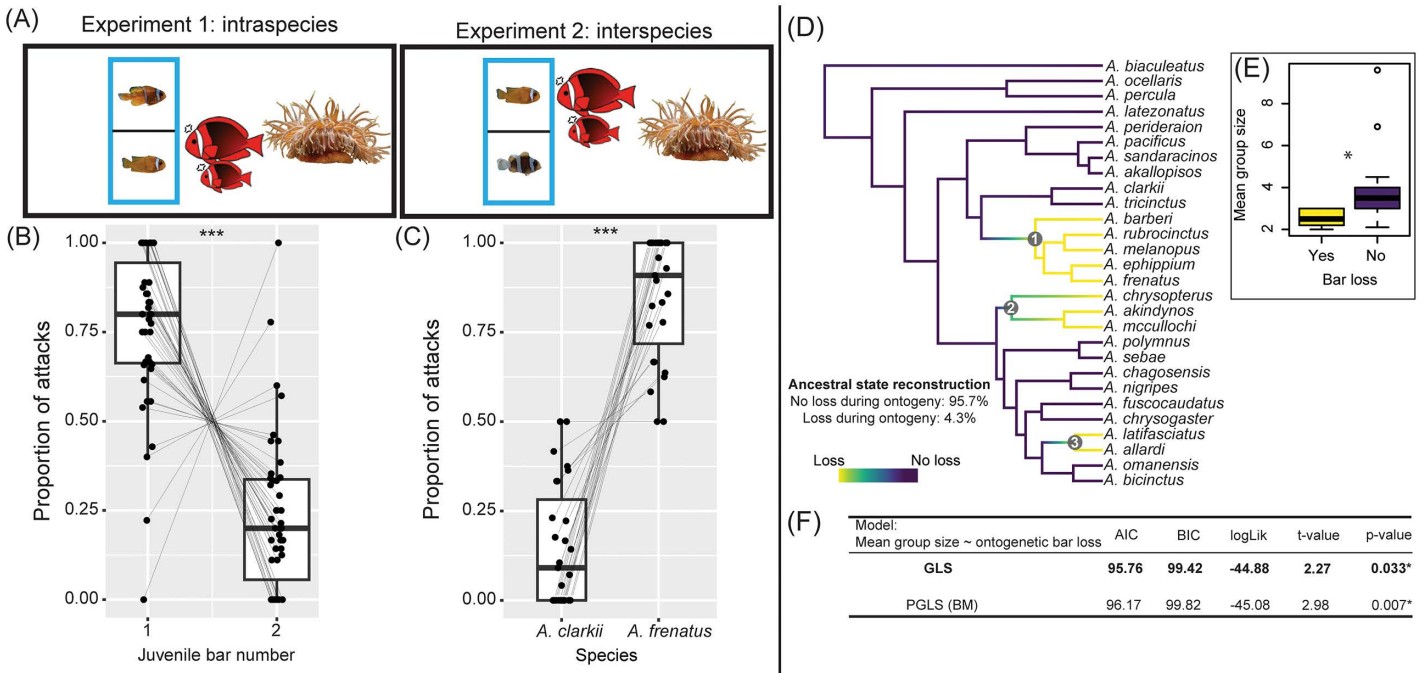

**Fig 6. Transient barring has a social signaling function. (A)** Behavior experiments compared the agonistic behaviors of adult *A. frenatus* (*n* = 3 pairs) towards different tested combinations of juveniles. **(B)** Proportion of attacks made during 10-minute trials by adults towards one bar and two barred *A. frenatus* juveniles (in Experiment 1), and **(C)** toward juveniles of *A. frenatus* and *A. clarkii* with one and two bars, respectively (in Experiment 2). Points connected by lines represent individual fish and trials, respectively. **(D)** Anemonefish species phylogeny from Gaboriau and colleagues [30] depicting the three clades which feature ontogenetic bar loss. **(E)** Average group sizes calculated per species and grouped by the presence (yes/no) of bar loss during ontogeny. Group size data were collected from primary literature or curated iNaturalist observations (*N* = 1,018 observations, with an average of 40 observations per species). **(F)** Summarized results from GLS and PGLS models analyzing the mean group sizes of species according to the status of bar loss during ontogeny. Box plots show the median, 25th and 75th percentiles, and range (whiskers). "*" = *p* < 0.05; "***" = *p* < 0.0001. The data underlying this Figure can be found in https://doi.org/10.5281/zenodo.17973175.

by adults directed between conspecific juveniles (one bar, *n* = 18) and the size-matched juveniles (ΔSL = 1.59 ± 1.10 mm) of a darker skin toned congener, *A. clarkii* (two bar, *n* = 18). Adults directed on average about four times more attacks towards conspecific juveniles (5.11 ± 0.74) than *A. clarkii* (1.30 ± 0.22) (binomial GLMM, *n* = 46, estimate = 2.56, SEM = 0.31, *z* = 8.39, *p* < 0.0001; Fig 6C, S5 Table). Together, it seems that adult tomato anemonefish used bar number or white skin area as a means of species recognition, where a single head bar elicited the highest agonistic response.

These behavioral results clearly suggest that the difference of white bar number (or total area) is highly important for juveniles. Counterintuitive to our above results, those with one bar were more likely to be attacked than two barred individuals. Thus, it appears that this developmentally plastic trait holds ecological significance, but its functional importance may be limited to during recruitment.

## Ontogenetic bar loss is associated with smaller social groups in *Amphiprion*

To understand how transient barring more widely relates to ecological traits in anemonefishes, we re-analyzed the evolution of ontogenetic bar loss across 28 species using the most-comprehensive anemonefish phylogenetic tree to date [30]. Ten species exhibit bar loss during ontogeny which form three distinct clades (Fig 6D). Ancestral state reconstruction analysis indicated a high likelihood (~96%) that ontogenetic bar loss did not feature in the last common ancestor of all anemonefishes, which is in line with an ancestral three barred form [28].

We then analyzed ecological factors (group size, number of host species) that might have contributed to the convergence of bar loss during ontogeny using both phylogenetic and non-phylogenetic least squares analyses (PGLS/GLS). Both linear models, even if the GLS model should be favored according to lower BIC and AIC scores, revealed that species with bar loss during ontogeny are associated with having larger mean group sizes (mean ± SE = 2.53 ± 0.12, $n$ = 10) than species without bar loss (3.89 ± 0.50, $n$ = 15) (Fig 6E and 6F). Similar to the results of Salis and colleagues [28], no differences in the total number of associated host anemone species or reproductive hosts (as defined by Gaboriau and colleagues [30]) were detected between anemonefishes showing ontogenetic bar loss and anemonefishes without bar loss. These macroevolutionary analyses indicate that bar loss during ontogeny tends to occur in anemonefishes that live in smaller group sizes.

## Discussion

Developmental plasticity or the capacity of a genotype to give rise to alternative phenotypes in response to varying conditions, is core to Eco-Evo-Devo research and plays a pivotal role in shaping the many traits that organisms express [56]. Elucidating the cellular and molecular processes underpinning developmental plasticity is therefore essential for understanding how novel adaptive traits emerge [57]. Here, we revealed environmentally induced temporal plasticity in the tomato anemonefish (*A. frenatus*) that exhibits a convergent, juvenile-to-adult reef fish color pattern transition. Specifically, the loss of the white body bar during ontogeny in *A. frenatus* occurred earlier, independent of body size, when in the presence of adults (Fig 7A and 7B). This transition from white to orange skin was facilitated by the mass apoptosis and halted production of iridophore cells. Additionally, increased TH production-related gene expression and activity in the skin might form a neuroendocrinal link between the perception of social cues and shifts in chromatophore populations. Transient barring appears to have a social signaling function, as adults were less agonistic towards two barred juveniles than the adult (one bar) phenotype. Combined with the prevalence of transient barring in species with smaller group sizes, it could mitigate conflict between recruit-stage juveniles and adults where there is less social buffering (i.e., fewer juveniles). These results provide a compelling example of how environmentally sensitive developmental mechanisms can drive ecologically significant phenotypic variation.

### Decreased survival and proliferation of iridophores underly bar loss

Examining TEM images taken in the body bar revealed that underlying the loss of white skin was a shift from predominantly structural-based coloration (iridophores) to pigment-based coloration (melanophores and xanthophores). Interestingly, no evidence was found for the dispersal of iridophores into the surrounding orange skin, nor the obscuring of iridophores by overlying pigmentation. Similarly, there were no traces of intermediate cell types indicative of cell transdifferentiation (e.g., melanophore-leucophore conversion in zebrafish [58]). Rather, both qualitative and quantitative evidence supported the mass cell death of iridophores. Although apoptosis is widely believed to be involved in color pattern change, its direct contribution has rarely been demonstrated (but see chromatic adaptation in zebrafish [59] and medaka [60]).

Iridophores increasingly exhibited symptoms of apoptosis with bar loss, including dark patches of condensed nuclear chromatin at the margins of nuclei, shrunken cell size, convoluted cell shapes, and eventually broken nuclei (karyorrhexis) and/or small, fragmented apoptotic cell bodies (apoptosomes) [61,62]. Moreover, we observed that some cells had apparently been displaced into the collagen layer above their primary site at the dermis-hypodermis boundary. Cellular detachment from the extracellular matrix by a loss of focal adhesion (i.e., anoikis) is another characteristic of apoptosis [61], and is promoted by Jun N-terminal kinase (JNK) in the presence of caspases (e.g., CASP3) [63]. Both genes, among other pro-apoptotic genes (e.g., TP63, SIK1, NOXA), were upregulated in fading white skin. Indeed, apoptotic activity peaked at the receding edge of the white bar, as confirmed in-situ by TUNEL assay and the IHC labeling of cells using cleaved CASP3 antibody. Most convincingly was the in-vivo demonstration that bar loss was pharmacologically impaired

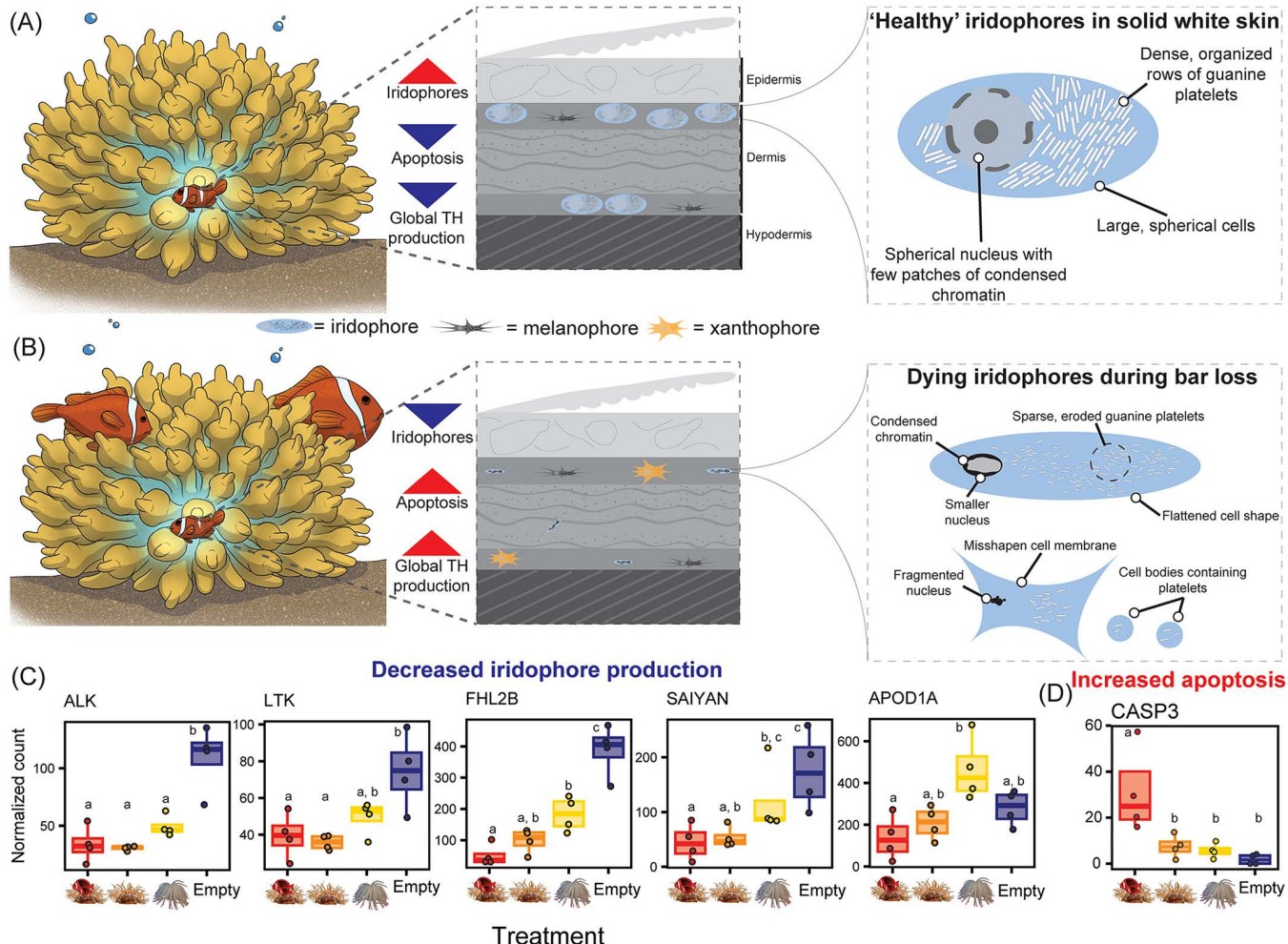

**Fig 7. Socially mediated, developmental plasticity of white bar loss. (A)** Loss of the white body bar was most advanced in juveniles which cohabitated with a pair of adults than **(B)** without adults. This coincided with drastic changes in iridophore cell abundance and morphology consistent with a shift towards a dying cell population. **(C)** Bar loss coincided with decreased iridophore-related gene expression (normalized gene counts), and **(D)** increased pro-apoptotic gene expression, especially Caspase-3. Letters denote statistical significance grouping ($p < 0.05$). Illustrations of *A. frenatus* and sea anemones are used with permission given by the artist, Rachel C. Vella. The data underlying this Figure can be found in https://doi.org/10.5281/zenodo.17973175.

by treating fish with caspase inhibitor (Z-VAD-FMK); however, its persistence, albeit diminished, suggests either inefficient (oral) drug delivery or caspase-independent pathways of cell death acted in-parallel (e.g., calpain-mediated apoptosis, autophagy) [64].

Bar loss also corresponded with the downregulation of iridophore gene markers (Fig 7C), possibly due to apoptosis, including *alk*, *apod1a*, *fhl2b*, and *saiyan* [29]. Like zebrafish, the false clownfish (*A. ocellaris*) depends on anaplastic lymphoma kinase and leukocyte tyrosine kinase (ALK/LTK) signaling in iridophore formation, proliferation, and survival [29,65]. Additionally, *saiyan* and *fhl2b* are essential for normal iridophore cover in zebrafish skin [29], the latter also being implicated in egg-spot formation in haplochromine cichlids [42].

TH, known to control bar formation in *A. ocellaris* [20], may also regulate ongoing pattern maintenance, as bar loss follows the reverse (posterior-anterior) order of their formation. Our data suggest that social context modulates TH

signaling in *A. frenatus*. Juveniles cohabiting with adults showed a downregulation of *dio3a*, implying reduced local TH inactivation, and upregulation of *klf9*, a TH target, supporting increased TH activity. These shifts indicate that social interactions in anemones elevate TH signaling in the skin. The TH receptor *trβ* was downregulated during bar loss (Fig 5A), possibly reducing cellular sensitivity to TH [66]. Although this contrasts with a general rise in TH levels, TH action is often cell-specific [67]. Given TH's known role in tissue remodeling, including in anemonefish metamorphosis [20,37], altered TH signaling likely promotes iridophore apoptosis and dermal reorganization. Although these transcriptional changes are consistent with shifts in TH signaling, these findings remain correlative until TH levels are directly measured. Further work should also dissect the neuroendocrine basis of bar loss, including species differences such as complete bar loss in *A. ephippium* versus caudal-only loss in *A. chrysopterus*, and the role of TH gradients along the body axis.

Interestingly, many pigment cell-related genes were downregulated during bar loss despite the shift towards pigment-based coloration. One of these genes was *oca2* (oculocutaneous albinism type 2), which is crucial in melanin synthesis and, in its defective state, is linked with hypopigmentation in vertebrates [47–50]. Potentially contributing to the disruption of the white-orange (black) skin boundary was a breakdown in cell-cell communication between chromatophores, a crucial factor in maintaining the form of many color patterns [68]. One candidate gene possibly involved was *gja5b*, an ortholog of the zebrafish *leo* gene, which encodes the 41.8 gap junction (CX41.8) [52,53] and instructs the distribution of iridophores in zebrafish stripes [52].

## Developmental timing of bar loss was influenced by social conditions

Bar loss in *A. frenatus* has a variable trajectory during ontogeny that is highly plastic and was advanced by ~24 days in the presence of adults. This finding was supported by observations on the reef, which related the one bar phenotype with the presence of congeners. Plasticity in color pattern development in response to social conditions has been reported in other animals, such as advanced egg-spot formation in socially-isolated haplochromine cichlid [69], territorial (yellow) and non-territorial (blue) African cichlid coloration [70,71], and delayed plumage coloration in socially-isolated zebra finches [72]. Repeated social domination can induce excessive cellular stress that promotes internal injury and the overexpression of pro-apoptotic factors (e.g., in mice [73,74]). Juveniles in our experiment were frequently harassed by adults that could have elevated stress and pro-apoptotic gene expression. Few reports of stress-related changes to structural coloration exist concerning dynamic color changes by alterations in iridophore spacing and crystal platelet organization, e.g., in zebrafish [75], frog (*Hyla* spp.) [76], and lizard (*Ctenophorus decresii*) [77]. Potentially mediating between sensing the social environment and cell stress was TH signaling and/or glucocorticoids, both of which respond to social stress [38,39] and can have interacting effects [78]. Future investigation into the role of stress in bar loss should compare the effect on juveniles when exposed to isolated adult visual cues (i.e., able to see but not interact with adults) to reveal whether physical harassment triggers accelerated bar loss.

Notably, the interaction with a host sea anemone also advanced bar loss. Despite the absence of a host anemone being an unnatural scenario, this demonstrated another example of the influence which the sea anemone symbiosis has on anemonefish development and the importance of its consideration in experiments. The symbiosis with sea anemones has been a major driver of adaptive radiation in anemonefishes [27] and the convergence of color patterns [30]. Moreover, anemonefishes can exhibit plasticity in their degree of melanization depending on the host species [20,79]. Aside from an elevated expression of some pro-apoptotic genes (e.g., CASP-3, JNK, CAPN1), there was little distinction in the skin transcriptomic effects of adult cohabitation when controlling for the presence of a live anemone. Although major transcriptomic differences might exist in other tissues/organs (e.g., brain, liver) linked to the discrepancies in bar loss, its consistent transition in appearance suggests a common underlying control mechanism with differences in its onset and/or activity.

One limitation in our experimental design was the lack of replicating the adult occupancy without a live host anemone. Thus, we could not fully separate the interacting effects of a host and social conditions on bar loss. Future work isolating the effects of sociality from symbiosis could aid in fully characterizing the mechanistic basis of the developmental

plasticity. However, this comparison would provide little-to-no ecologically relevant insights into the plasticity of bar loss, as all anemonefishes are obligatory symbionts meaning that the local host and social conditions are inextricably linked. Additionally, small and/or poor-quality hosts can limit group size and recruitment [80,81], as dominant anemonefish increasingly forcibly evict subordinates or reject recruits [82]. The heightened risk of adults in artificial habitats attacking juveniles raises practical and ethical concerns.

### Species recognition based on bar patterning and the benefit of transient bars

Our finding that adult *A. frenatus* were more aggressive to one barred juveniles is in-line with the idea that anemonefishes use the number of white bars (or white skin area) for species recognition, as shown in *A. ocellaris* [34]. This ability to distinguish conspecifics based on the patterning of white skin combined with interspecies variation in color pattern appears to be important in limiting competition and facilitating mixed species coexistence within sympatric anemonefish communities [28]. Our results support the notion that this species recognition system is a general aspect of behavioral ecology in anemonefishes.

Intriguing is the notion that transient bars might function as a recruitment aid in some anemonefishes. Other reef fishes exhibit "non-threatening" color patterns as juveniles to aid in settlement by mitigating adult aggression, e.g., emperor angelfish (*Pomacanthus imperator*) [21] and some damselfishes (Pomacentridae spp.) [22,83,84]. Similarly, the multiple white bars of *A. frenatus* juveniles may confer a recruitment advantage by appearing distinct from the adults, while the one bar morph is more recognizable as a conspecific. Distinct ultraviolet/UV coloration can also signal subordinance in juveniles of the Barrier Reef anemonefish (*A. akindynos*) [32]. Both the ability to produce and visually perceive UV appears throughout anemonefishes [85–87]. Because juveniles in our behavior experiments were in UV-transmissive containers (S8 Fig), it is possible that a boosted UV signal from a second bar could have amplified a non-threatening appearance.

Temporal plasticity in bar loss provides a flexible phenotype to suit variable social contexts at anemones. Recruit-stage juveniles with transient barring can either settle into vacant anemones or join adults and/or (older) juveniles. For lone settlers, it may be a safer strategy to delay bar loss and avoid potential displacement by invading adults, especially in the presence of neighboring colonies [88,89] or during habitat loss (e.g., from typhoons or anemone bleaching from marine heatwaves) [90,91]. Counterintuitively, we found that the body bar was more rapidly lost in the presence of adults; however, transient barring could fulfill a short-lived function only during recruitment. Post-recruitment, it may be more beneficial to adopt the adult pattern to be recognized as a conspecific and participate in the social hierarchy, and/or possibly for camouflage. Indeed, the idea that the adult color pattern transition is related to social rank ascension has previously been posed for *A. frenatus* and other anemonefishes [91]. Long-term, in-situ assessments are needed to verify whether transient barring confers any benefit to recruitment. Further support that transient barring is a socially important trait came from our phylogenetic analysis, which revealed its prevalence in species living in smaller colonies containing, on average, 1.4 fewer ranks. While we did not find evidence of bar loss during ontogeny being linked to host specificity or the number of host associations, it remains to be examined whether preferred host traits (e.g., area, coloration, toxicity, tentacle length) contributed to its evolution, as has been suggested for other anemonefish color pattern traits (see [30,35]). Smaller groups have less social buffering capacity than larger groups, where aggression is diffused among more members [92]. Less social species like *A. frenatus*, typically have groups containing only one to two juveniles that are extremely small in size compared to adults [25]. Thus, transient barring could help in avoiding potentially lethal conflict during the recruitment phase.

## Conclusions

Our study clearly highlights how social interactions can profoundly influence developmental outcomes through endocrine mechanisms. Our data suggests that TH, traditionally known for the control of growth, metabolism, and metamorphosis, may also act as a systemic mediator linking social context to plastic pigmentation changes in a coral reef fish. More

generally, our results illustrate how environmental and social cues can become deeply integrated into the developmental regulation of adaptive traits. Such processes shape both individual phenotypes and species-level diversity to directly contribute towards the dynamic and context-dependent nature of biodiversity in natural ecosystems.

## Materials and methods

### Animals

**Ethics approval.** All experiments and field work were conducted with prior approval from the Animal Experiment Regulations at Okinawa Institute of Science and Technology (OIST) Graduate University (Approval No. ACUP-2024-014; ACUP-2023-012) and adhered to the national Basic Guidelines for Animal Experimentation established by MEXT.

**Livestock upkeep and rearing.** Larvae of *A. frenatus* and *Amphiprion clarkii* were obtained from two breeding pairs and one breeding pair, respectively. The mariculture of larvae followed established protocols as described in Roux and colleagues [93]. Adults and juveniles older than 18 days-post-hatching (dph) were maintained at the OIST Marine Science Station in outdoor flow-through systems that received natural sea water. Broodstock regularly spawned approximately every 2 weeks, and in total, four egg clutches were hatched and raised to the juvenile stage (14 dph and older). Host sea anemones (*Entacmaea quadricolor*, $N = 3$) used in this study were originally wild caught from Okinawan reefs.

## Method details

### Field observations

Anemonefish colonies were observed at multiple coral reefs by scuba in Okinawa, Japan (*A. frenatus* at Ibu Beach, Yomitan), French Polynesia (*A. chrysopterus* at Bora-Bora), and northern Barrier Reef, Australia (*A. melanopus* at Lizard Island). Juveniles were observed either alone or cohabiting with conspecifics at isolated anemones (containing one or more adult or juvenile conspecific) or in colonial groups, where multiple breeding pairs and juveniles existed within a large area (6–12 m²) containing numerous anemones. Juveniles were recorded for their bar pattern (1, 2, or 3 bars) and closely monitored for 5-min to observe and record any cohabiting individuals within their anemone.

### Developmental bar loss experiment

A developmental experiment was conducted to assess the effect of different environmental conditions on bar loss. Experiments were run in three serial replicates (between 24/07/2023–6/09/2023; 6/09/2023–20/10/2023; and 28/11/2025–18/12/2025) using three clutches of 18dph juveniles ($N = 76$) produced by a single breeding pair. Any juveniles that exhibited misdeveloped (incomplete or asymmetrical) barring were excluded. Eight juveniles were allocated to four outdoor tanks (90 × 60 × 60 cm) including: (1) containing an anemone (*Entacmaea quadricolor*), (2) an anemone (*E. quadricolor*) with two adult fish (Pair/Replicate 1: male 1 = 45.08 mm, female 1 = 74.29 mm; Pair/Replicate 2: male 2 = 41.86 mm, female 2 = 71.99 mm), (3) two plastic anemones (Mallofusa brand), and (4) an empty tank. Because of the fragility of juveniles and high risk of mortality, it was not possible to accurately measure the initial body lengths of fish assigned to each treatment using a microscope. Approximate measurements were instead taken from overhead photographs using a phone camera of the juveniles in a shallow dish containing a tape measure. There were small (<1 mm on average) differences among the initial standard lengths of juveniles in Replicate 1 (SL mean ± SD, Occupied anemone = 8.27 ± 0.45 mm; Anemone = 8.29 ± 0.52 mm; Fake anemone = 8.05 ± 0.67 mm; Empty = 7.92 ± 0.66 mm), Replicate 2 (Occupied anemone = 8.17 ± 0.61 mm; Anemone = 8.13 ± 0.65 mm; Fake anemone = 8.07 ± 0.68 mm; Empty = 8.15 ± 0.84 mm), and Replicate 3 (Occupied anemone = 9.17 ± 0.71 mm; Anemone = 8.96 ± 0.63 mm; Fake anemone = 8.99 ± 0.67 mm). More accurate (microscope-taken) measurements of four randomly sampled (for RNA sequencing) pretreatment juveniles indicated there was reasonable uniformity among initial body lengths (Replicate 1 = 8.70 ± 0.15 mm; Replicate 2 = 7.85 ± 0.26 mm; Replicate 3 = 10.39 ± 0.56 mm). Experimental tanks were monitored intermittently (daily or

every second day) by direct observation and underwater video (GoPro Hero 5) for undisturbed monitoring. During the first replicate one and two juvenile(s) were found dead by unknown causes within the first 24-h in the Empty treatment and Anemone treatment, respectively. These individuals were replaced with spare siblings. Similarly, three juveniles in the second replicate died in the plastic anemone treatment and were replaced within 24-h of commencing the experiment. Because the third replicate had a very low larval rearing survival, it was decided to limit the experiment to three treatments and exclude the Empty treatment. Each treatment was fed one to two times per day with pellet food (Hikari). Sea anemones were directly fed frozen shrimp once per week. Subsamples of juveniles ($n = 4$) were sampled at Day 20 (38 dph), and the remaining individuals were sampled at Day 44 (62 dph). However, the third replicate was terminated on Day 20 due to multiple mortalities in all three treatments (Occupied anemone, $n = 5$; Anemone, $n = 3$; Fake anemone, $n = 5$). Sampled fish were euthanized by exposure to MS222 (Sigma-Aldrich; 200 mg/L) in sea water and immediately photographed on both sides under a Zeiss stereomicroscope (V12 Discovery) using an Axiocam 105 camera and illuminated by white LEDs (Zeiss CL6000). The body bar regions (both sides) of juveniles were then dissected by making two straight dorsoventral cuts in line with the eighth dorsal spine and the fourth dorsal ray. Body bar samples were cleaned of bones and any attached organs, then stored in RNAlater (ThermoFisher Scientific, Waltham, USA) at 4°C for 24-h and −80°C thereafter.

A separate cohort of 10 juveniles were reared (from 15/08/2024 to 4/09/2024) in sea anemones either with or without adults (Pair 1) for whole body transcriptome sequencing. Juveniles ($n = 5$) were held in one of two environmental treatments for 21 days (23 dph till 44 dph), then euthanized and preserved in RNAlater.

## Resin embedding and transmission electron microscopy

To examine changes in chromatophore morphology and populations during bar loss, we used TEM on cross-sectioned skin of juvenile *A. frenatus* ($N = 6$) that exhibited varying degrees of bar loss. Fish were first euthanized (MS222, 200 mg/L) and photographed on their side. Body bar skin and adjacent orange skin were extracted using disposable biopsy punch pens (5.00 mm diameter) and trimmed into ~2 mm wide pieces (Fig 3B). Tissue fixation, staining, and preparation (dehydration steps and resin embedding) followed a standard 3-day TEM procedure, as per Miyake and colleagues [54]. Resin blocks containing fixed tissue were roughly cut and trimmed using a razor blade and then finely trimmed using a 45° 2.0 mm diamond knife (Diatome) and an ultramicrotome (Leica EM UC7) until the sample was reached. Multiple thin 90 nm sections were cut and loaded onto hexagonal 100 mesh copper grids (Electron Microscopy Sciences).

Grid-loaded sections were imaged using an electron transmission microscope (JEOL1400 Flash, HT voltage = 100 kV, beam current = 53–55 μA). A widefield overview of the skin layers was imaged at x800-1k mag., individual chromatophores in skin were imaged at x1.5k mag., iridophore nuclei and guanine platelets were imaged at x8-10k mag. Multipanel wide-angle images were stitched together using the pairwise stitching plugin in FIJI/ImageJ [94]. Chromatophore cell counts and individual cell areas were measured by using the cell counter and manual tracing of cells. The aspect ratios of iridophore guanine platelets ($N = 378$) were calculated using measurements of the longest and shortest dimensions of individual platelets.

## Transcriptomic analysis

**RNA extraction.** Four juveniles per environmental condition ($N = 32$) were sampled for skin transcriptomes. Additionally, five juveniles were sampled for performing whole-body RNA extractions that were raised either with or without an adult pair. Skin with attached muscle RNA and whole fish RNA were extracted using a custom TRIzol protocol and Promega Maxwell RSC simplyRNA Tissue Kit (AS1340, following the manufacturer's protocol), respectively. Skin tissue samples were first washed in 1.0 mL of TRIzol reagent (Ambion Life Technologies) to remove excess RNAlater. After tissue was lysed/homogenized (MP FastPrep-24 5G) and then centrifuged (5-min at max. speed) at 4–10°C. Lysed samples were then incubated for 5-min to allow the complete dissociation of nucleoprotein complexes. Next, samples were treated with 0.2 mL of chloroform, incubated for 2–3 mins, and centrifuged for 15-min (12,000$g$ at 4°C). The

colorless aqueous phase containing RNA was then transferred into new tubes. An equal volume of RNase-free ethanol was then slowly added and gently mixed by pipette. Samples were loaded onto RNeasy MinElute spin columns in a 2 mL collection tube and centrifuged for 15-s at 8000$g$. Followed by two separate buffer washes, RW1 (700 µL) and RPE (500 µL), centrifuged for 15 s (at 8000$g$), then a 80% ethanol (500 µL) wash centrifuged for 2-min. Spin columns were then transferred into new tubes and centrifuged at full speed for 5-min to dry the membrane. RNA was then eluted into 1.5 mL tubes by loading membranes with 50 µL of RNase-free water, incubated for 5-min and centrifuged (at 8000$g$) for 1-min. RNA-quality was checked prior to sequencing using an RNA ScreenTape Assay (Agilent TapeStation 4200) that indicated good quality RIN$^e$ scores ≥6.6.

**RNA-seq libraries preparation and sequencing.** RNA-seq library preparation and Illumina sequencing were performed by the OIST Sequencing Section (SQC). RNA-seq libraries were produced using the NEBNext Ultra II Directional RNA Library Prep Kit for Illumina. The pooled library was then split into two Illumina Nova Seq 6000 S4 flow cell lanes for sequencing with 150 bp PE reads.

## Cryosectioning and TUNEL assay

To assess whether apoptosis was elevated in fading white skin, a TUNEL assay was performed using one juvenile (ID: TA1, ~1-month old) with a partially faded white bar and an older juvenile (ID: TA2, ~3-months old) with a heavily faded white bar. These were non-experimental fish from a separate clutch (but same parents) as those used in the development experiment. Fish were euthanized (MS222, 200 mg/L) and then had their left-side body bar regions dissected (TA1 = 7.00 × 1.30 mm; TA2 = 10.20 × 1.63 mm), embedded in cryopreservation medium (SAKURA: Tissue Tech O.C.T. Compound), and snap frozen using dry ice. Samples were stored in −80°C until use. A Cryostat PHC Cryostar NX70 with a microtome blade was used to make transverse 10 µm thin cryosections that were transferred using a brush onto pre-cooled glass slides (Matsunami micro slide glass, 76 × 26 mm). Sectioned regions included the ventral (orange), middle (faded white/orange), and dorsal skin (solid/faded white) (Figs 4B and S4). Including both TUNEL-treated and controls, there were nine sections per region ($N$=27) made in TA1, while two sections per region ($N$=6) were made in TA2. Sections affixed to slides were air-dried and stored in a dark box at −80°C until use.

A commercial TUNEL assay kit was used (Roche In situ cell death detection kit, fluorescein, cat no. 11684795910) as per the manufacturer's instructions for cryopreserved tissue sections. Sections were then labeled as per the adherent cells protocol, including the preparation of TUNEL treatment, positive (nuclease S7 incubated), and negative control (excluding enzyme solution) sections. Slides were coverslip-mounted (Matsunami, 18 × 18 mm) using a DAPI-containing mounting medium (Vectashield H-1200).

Micrographs of sections were taken at x20/0.75 (UPlanSApo WD0.6) magnification using a spinning disk confocal microscope (Olympus/Evident SD-OSR CSU W1) and Prim BSI camera. For fluorescent images, we used an excitation and detection wavelength of 488 nm and 561 nm, respectively. TUNEL assay section micrographs were observed in imageJ [95] and manually outlined (fluorescent) +TUNEL cells to record set measurements, including "area," "area integrated intensity," and "mean gray value." In addition, we measured mean background fluorescence in an adjacent large area free of +TUNEL cells.

## Paraffin embedding and immunohistochemistry

Two fish, one with partial bar loss (ID: CA1) and another with complete bar loss (ID: CA2), were euthanized (MS222, 200 mg/L), photographed on their sides, and had their body bar regions dissected (CA1 = 7.00 × 1.10 mm; CA2 = 9.00 × 1.20 mm). Skin fixation, dehydration, clearing, paraffin embedding, and deparaffinization steps were followed, as previously described by Miyake and colleagues [54].

Prior to staining, the unmasking of antigens was performed by microwaving sections/slides at sub-boiling temperatures (95–98°C) for ~10-min in 1X citrate unmasking solution (SignalStain) and then cooled at room temperature for

30–40 min. The staining procedure followed the antibody manufacturer's protocol for immunohistochemistry (Cell Signaling Technology). In short, sections were incubated with 400 µL of blocking solution (1X Animal-Free Blocking Solution, Cell Signaling Technology #15,019) for 1-h at room temperature. After removing the blocking solution using a dH$_2$O bath, the sections were incubated in 400 µL of cleaved caspase-3 primary antibody (Cell Signaling Technology Cat# 9,661, RRID:AB_2341188) diluted 1:400 with antibody diluent (1X PBST/5.0% normal goat serum) (Thermo Fisher Scientific Cat# 31,872, RRID:AB_2532166) overnight at 4°C. Control sections were incubated in 400 µL of antibody diluent without primary antibody. During the next day, sections were removed of antibody solution with three washes using TBST (1X Tris-buffered saline with Tween 20) for 5-min each. Sections were covered with 1–3 drops of SignalStain Boost Detection Reagent (Cell Signaling Technology Cat# 8125, RRID: AB_10547893) and incubated in a dark, humidified chamber for 30-mins at room temperature. Next, sections were exposed to 400 µL of (30 µL) SignalStain DAB Chromogen Concentrate mixed in (1 mL) SignalStain DAB diluent and incubated at room temperature (8-mins). After washing sections in dH$_2$O, the slides were immersed for 10-min in hematoxylin as a nuclear stain then rinsed of excess stain using water. Post-staining, the sections were rapidly dehydrated through ascending ethanol concentrations lasting 1-min each (70%, 80%, 90%, 95%, and ×2 100%), cleared using Lemozol (x2, both 5-mins) and xylene (5-mins), and coverslip-mounted using mounting medium (Sigma-Aldrich, Entellan new).

Stained sections were observed under a light microscope (Zeiss Axiocam 705 color camera connected to a Zeiss Axio Imager 2) at 4× and 20× magnifications. Micrographs were then observed using ImageJ [95] for the presence of brown +DAB staining indicative of caspase-3 activity.

## Pharmacological experiment

To attribute apoptosis to bar loss, we pharmacologically blocked caspases in-vivo and compared white skin area after 26-days in two serially run replicates (between 26/11/2024–20/12/2024 and 6/01/2025–31/01/2025). Both replicates used a single cohort of juveniles (hatched on 1/09/2024) with eight individuals assigned per treatment/control group ($N = 48$). Juveniles were fed two times per day with ~11 mg of pellets, an average food intake calculated beforehand using equivalent-sized fish ($10.90 \pm 5.01$ mg, $n = 5$). Pellets fed to juveniles in two treatment groups were presoaked in either 1 mg/mL (low dose) or 2 mg/mL (high dose) of broad caspase inhibitor Z-VAD-FMK (Selleck Biology, CAS No. 187389-52-2) dissolved in 1X DMSO. The control group was fed pellets presoaked in DMSO. Drug and DMSO-soaked food were prepared in advance in 1-week batches that were mixed, air dried in a cold room, then weighed out into 11 mg portions and stored in PCR strip tubes at 4°C until use. The low dosage was based on that reported in the literature for intravenous delivery in adult zebrafish [96] and doubled (high dosage) to account for the less efficient (oral) delivery and larger size of anemonefish. Note that a triple dose (3 mg/mL) was trialed but found to be lethal with 90% mortality from one-time feeding. In addition, one juvenile (ID: HT9) from the double dosage group died from an unknown cause a day before the termination of the second replicate. Juveniles were individually housed in plastic screw-top containers (500 mL, 90.00 mm diameter) drilled with numerous holes (5 mm) and floated in one of three closed systems (600 × 300 × 300 mm). In each system, water temperature was kept at 27.0°C, salinity between 33‰ and 34‰, and pH between 8.1 and 8.2. Water parameters were maintained by a sponge filter and exchanging 1/3 total water volume with clean artificial sea water every 2 days.

Prior to the experiment, juveniles were anaesthetized in 100 mg/L of MS222 and quickly photographed on both sides while submerged in a white tub alongside a clear resin-embedded color standard with scale bar (Calibrite Colorchecker Classic). Photographs were taken from directly (17 cm) above submerged fish using a camera (Olympus OM System TG-6) mounted on a stand (SFC SL700) with illumination provided by two side-mounted white LED lamps (Grangrade 12.6W LDG12D-G AG6/RA93). An identical camera setup and settings (+1.3 ISO, x2.0 zoom) were used for taking post-experiment (Day 26) images. All juveniles were euthanized (MS222, 200 mg/L) upon terminating the experiment.

## Behavioral experiments

To test whether adults could distinguish juveniles based on white barring, we compared adult responses towards juveniles with one or two bars. Experimental trials used three adult pairs, including the same two pairs from the developmental experiment (Pairs 1 and 2) and a new third pair (Pair 3: male = 43.34 mm, female = 81.86 mm). Experiment 1 trials ($n$ = 49) used 2–5 month-post-hatching *A. frenatus* ($N$ = 38) with/without bar loss. Experiment 2 trials ($n$ = 46) used one bar *A. frenatus* juveniles ($n$ = 20) and *A. clarkii* juveniles ($n$ = 20) that were equivalent in age. Fish were measured for their standard length using a measuring container (with ruler) and sorted into size-matched pairs that exhibited one bar and two bars. These pairs were separately held within clear plastic boxes (Big Fish House, 308 × 145 × 165 mm). Juveniles were prevented from interacting with each other by an opaque plastic divider blocking the view between chambers. Boxes containing juveniles were placed into adult tanks, weighed down by a dive weight, and centrally positioned ~100 mm away from the anemone. The sides of boxes were blacked out using an opaque sheet of corrugated plastic and were undisturbed for 10-min to allow juveniles to acclimatize before trials. Upon trial commencement, an underwater camera (Go Pro Hero 6) was positioned to record through the box towards the anemone (and adults), and the black sides were removed. Adult-juvenile interactions were recorded for 10-min after which the trial was terminated and reset using another pair of juveniles. The box position (left or right) of juveniles was switched each trial. There was no repeated testing of individual juveniles per adult pair.

Footage was analyzed by manually counting the number of aggressive behaviors directed at juveniles per adult, including attempts at biting, charging, and gaping. Other recorded variables included video/trial ID, juvenile IDs, tank side (left or right) of each juvenile, species (for Experiment 2), juvenile standard lengths (mm), calculated difference in standard length between juveniles (delta_SL), and bar number. Only in one trial did adults (Pair 1) completely ignore both juveniles, and was omitted from the analysis.

## Spectral transmission measurements

Because UV-signaling is important in anemonefish, we checked whether juvenile UV skin colors were visible to adults by measuring the spectral transmission of the clear plastic containers. Light emitted through a 200 μm (UV-VIS) fiber optic cable connected to a pulsed xenon lamp (Ocean Optics, PX-2) was measured through plastic using a spectrometer (Ocean Optics, USB4000) fitted with a bare (400 μm) fiber. The average normalized spectral transmission was then calculated from three measurements.

## Ecological data collection

To analyze the ecological traits corresponding with the convergence of bar loss, we compiled data on anemonefish host use and group size. Information was collected from the primary literature on the status of bar loss during ontogeny ("1" = present, "2" = absent), and number of host anemone associations. Host associations were defined as per Gaboriau and colleagues [30], including total number of host species associations and reproductive host use (i.e., anemone species known to host breeding pairs of anemonefish). Mean group size data were collected using either primary literature reports or, in their absence, scientific-grade observations from the publicly accessible database iNaturalist [97]. Individual counts of group members per colony were restricted to high-quality photographs showing the whole host anemone with at least two anemonefish. Our survey of group size was also limited to single colony scenarios and excluded multigroup/pair scenarios to avoid ambiguity with the social grouping of fish. As many suitable photographs as available per species were included in the survey that ranged from 5 to 110 observations (mean ± SE = 41.0 ± 8.0, $n$ = 25 species). Three species (*A. chagosensis*, *A. omanensis*, *A. pacificus*) were dropped from the analysis due to having fewer than three observations.

## Quantification and statistical analyses

### Statistical software

All statistical analyses and plotting were performed using R Statistical Software (4.4.0) [98].

## Image analysis and white skin area comparisons

To accurately quantify the area of white skin, we analyzed right-side-up images (.JPG, 2080 × 1220) of juveniles using a combination of the R packages recolorize [99] and patternize [100]. Images taken in the pharmacological experiment were treated to an additional pre-processing step that calibrated (RAW) images in case of variable lighting using the "Generate Multispectral Image" function in the MICA toolbox [101] imageJ [95] plugin. This step used the normalized reflectance values of the six gray standards of the Colorchecker Color Chart (Calibrite) to produce normalized and linearized color images (.JPG). Images were then aligned using the "alignLan" function in Patternize using 11 fixed morphological landmarks (S9A Fig) registered as XY coordinates in ImageJ/FIJI [94]. A mask outline was applied to limit analysis to the body bar region (S9B Fig). Note that in the development experiment, a unique mask had to be made for pretreatment (18 dph) juveniles due to large changes in morphology. Aligned image raster objects were then converted to arrays using the "brick_to_array()" function for compatibility with recolorize. The colors of aligned image arrays were remapped to a combined four-color palette (gray/white, orange, yellow, brown) in LAB color space using the "recolorize2" and "imposeColors()" functions. The proportional area of white pixels out of total pixel area was then calculated per individual using the plotted color palettes. Additionally, for the pharmacology experiment, the proportional decrease in white skin was calculated by:

$$((pixels\_pre - pixels\_post))/(pixels\_pre) \tag{1}$$

where "$pixels_{pre}$" and "$pixels_{post}$" were the pre- and post-experiment white pixel counts, respectively.

To analyze the effect of environmental treatment on bar loss, we ran an ordered beta (ordbeta) family regression analysis with logit link function suited for proportion data with lower/upper bounds of zero and one [102] using the glmmTMB [103] R package. The ordbeta model included the proportion of white skin ("prop_white") as the response, while the environmental treatment ("A" = adult cohabitated anemone, "B" = anemone, "C" = fake/plastic anemone, "D" = Empty tank), sampled age in dph ("38", "62"), the interaction between treatment and sampled age, and body size (standard length in mm, "SL_mm") were held as fixed predictor variables. Replicate ("1", "2", "3") was included as a random effect to account for differences between cohorts. Multiple pairwise comparisons were then performed using the "emmeans" and "pairs" functions in the R package emmeans [104] to calculate estimated marginal means and Tukey adjusted $p$-values between treatment comparisons within the sampled stages.

Another ordbeta regression analysis (using the glmmTMB [103] package) with logit link function analyzed the effect of Z-VAD-FMK treatment on white skin loss. This ordbeta model included the proportional decrease of juvenile white skin ("prop_white") as the response, while the treatment/control group ("A" = high dose, "B" = low dose, "C" = DMSO control) and body growth in SL ("deltaSL") were held as fixed predictor variables. Replicate ("1", "2") was included as a random effect. A post hoc adjustment test for Tukey multiple pairwise comparisons between treatments/control was conducted (using the R package emmeans [104]). Note that equivalent results were obtained using a similarly parameterized quasibinomial family generalized linear model with a weighted response encoded as the pre-experiment white pixel area.

The quality of fit was assessed for both ordbeta models with residual diagnostics and plots generated using the "simulateResiduals" function in the R package DHARMa [105]. In neither case were issues regarding dispersion and heteroscedasticity detected.

To analyze how white skin loss was related to body growth by treatment, we performed separate linear regression analyses using the "lm" function in base R. This analysis included the proportion of white skin as the response variable and standard length in mm ("SL_mm") as the independent variable. Note that response variables were either square root transformed or log-transformed to meet the assumption of normality. Shapiro–Wilk tests (base R) checked the normality of the residuals.

The change in white skin area within each treatment of the pharmacology experiment was analyzed using exact Wilcoxon signed rank tests for paired data per treatment/control (using "wilcox.test" function in base R). Each test compared

the pre-experiment and post-experiment white pixel counts per fish. The assumption of symmetrically distributed paired differences was assessed and found to be unviolated using the "symmetry.test" function in the R package lawstat [106].

Analysis of how body size varied in the developmental experiment was performed using a two-tailed, one-way ANOVA per sampled age group (38 and 62 dph) and replicate (1 and 2) using the base R function "aov". Each ANOVA included measured juvenile standard length "SL_mm" as the response and environmental treatment as the independent variable. Shapiro–Wilk tests were used to check the normality of residuals, and Levene tests assessed the homogeneity of variance across groups (in R package broom [107]). Juvenile SL in 62 dph analyses were log-transformed to meet the normality assumption. The adjustment of $p$-values for multiple paired comparisons was by Tukey HSD (in base R).

### Transcriptomic analysis developmental experiment

**Pretreatment (read quality, quantification, filtering, normalization).** Raw read quality was assessed using FastQC quality control tool (v.0.11.9 [108]). The MultiQC tool (v.1.28 [109]) was used to compile and visualize reports from across all samples. Because no high-quality (chromosome-scale) annotated genome or transcriptome exists for *A. frenatus*, we downloaded transcript sequences and annotations for a close relative, *A. clarkii*, from NCBI (https://www.ncbi.nlm.nih.gov/datasets/taxonomy/80970/). Poor quality and adapter sequences were trimmed from both paired and non-paired reads using the Trimmomatic tool (v.0.39 [110]). Transcript abundance quantification via the pseudoalignment of trimmed reads against an *A. clarkii* transcriptome index was then performed using Kallisto (v.0.51.1 [111] with 36 samples and 100 bootstraps). Gene counts as total abundance files were then imported into R using the tximport package (v.1.0.3 [112] with settings type = "kallisto" and txout = "TRUE"). Genes with low (≤10) counts across samples were filtered out. Normalized gene counts were obtained by the median of ratios method and computing the effective library sizes using the function "estimateSizeFactors" from Bioconductor package DESeq2 (v.1.12.3 [113]). Raw count values were then transformed using a variance stabilizing transformation ("vst" function from DESeq2 with setting blind = TRUE).

**Sample correlation and principal component analysis.** The distribution of samples was visualized by sample correlation and PCA plots computed using the total normalized gene count data. Pairwise Spearman's rank correlation coefficients between samples were calculated using the "cor" function in base R. Correlation coefficients were then visualized by heatmaps produced using the "heatmap.2" function in the gplots [114] R package that reported z-scores. PCAs were computed on centered, unscaled counts using the "prcomp" function (in base R), and the PC scores of samples were plotted using the ggplot2 [115] package in R.

**Differential gene expression analysis.** Differential expression analysis was performed on filtered raw gene counts using DESeq2 [113], that applied the Wald test for significance of GLM coefficients. Shrunked log2 fold changes were applied using the "lfcShrink" function (from DESeq2). The default Benjamini–Hochberg method (FDR) of $p$-value adjustment was implemented to control the false discovery rate. Significant differentially expressed genes met a threshold of 1.5-fold (0.58 log2-fold) change and adjusted $p$-value < 0.05. Multiple paired contrasts were computed between sample groupings based on environmental treatment, contact/no-contact with a live anemone (PC2—grouping), and body size (PC1—grouping). The absolute log2 fold-change and adjusted $p$-values of the filtered significantly expressed genes were visualized in volcano plots generated using the Bioconductor package EnhancedVolcano [116].

Gene pathway enrichment analyses were based on the GO terms originally assigned to *A. clarkii* genes by Moore and colleagues [117] that relied on BLAST output. GO enrichment analyses were performed using the R package TopGO [118] (with nodeSize = 10) for Biological Processes using an *A. clarkii* gene2go annotation. Enrichment tests were performed separately for negatively and positively regulated genes using the 'runTest' function with default algorithm ("weight01") [118] and Fisher's test for significance. The top 1,000 enriched GO terms were retrieved for each treatment comparison along with their gene names/IDs, corresponding $p$-values, and then plotted using ggplot2 [115].

Specific gene sets of interest were compiled from previously curated lists of orthologous *A. ocellaris* genes with functions related to chromatophore production/activity (see Herrera and colleagues [119]). Genes were also identified using

BLAST searches using the gene IDs of the top 1,000 differentially expressed genes, and those contained in enriched GO terms related to apoptosis or cell death. ANOVA was conducted to compare the normalized counts of individual genes between all four treatments with $p$-value adjustment for multiple comparisons by Tukey HSD (in R package broom [107]). The differentially expressed genes contained in these lists were then visualized using normalized and centered gene counts in heatmaps (using gplots [114]) and/or (non-centered) boxplots (using ggplot [115]).

### Histological quantitative analyses

The relative abundance of different chromatophore cells identified in TEM images was calculated as the proportion of cells out of the total count of chromatophore cells across all imaged sections per bar loss stage/area (full, $n = 3$; fading–dorsal, $n = 2$; fading–ventral, $n = 3$; faded, $n = 3$). Chromatophore relative abundances were plotted as stacked bar plots using ggplot2 [115]. Square root transformed iridophore cell areas (in $\mu m^2$) were compared among bar loss stages using one-way ANOVA with Tukey HSD multiple comparisons $p$-value adjustment (using R package broom [107]). Similarly, the (log-transformed) aspect ratios of individual guanine platelets calculated by dividing the longest dimension by the shortest were compared among bar loss stages using one-way ANOVA with Tukey HSD test. The normality of residuals was assessed using Shapiro–Wilk tests (in base R) and residual plots. Additionally, Levene tests (using the R package broom [107]) checked the homogeneity of variance across groups.

To provide a background-independent measure of cell fluorescence intensity in TUNEL assay micrographs, the area CTCF was calculated per +TUNEL cell by

$$integrated\ density - (individual\ cell\ area\ \times\ mean\ background\ fluorescence) \tag{2}$$

### Behavioral data analysis

To assess the effect of juvenile bar pattern on the aggressive behavior of adults, we ran binomial family generalized linear mixed effects models (GLMMs) with log-link function using the "glmer" function in the R package lme4 [120]. Global GLMMs included the combined weighted proportion of attacks (attack count / total attack count) by adult pairs as a binomial response, while juvenile bar number ("1" or "2"), the scaled difference between juvenile standard lengths ("delta_SL" in mm), and left or right tank side of juveniles ("tank_side" as "L" or "R") were fixed predictor variables. The parameter "weights" was set as the total number of attacks. Both tank ID (i.e., pair ID) and individual trial ID were included as random effects to account for variability between adult pairs and trials, respectively. Because juveniles were presented as pairs in experiments, we avoided the pseudoreplication of trials by arbitrarily assigning a focal juvenile in each GLMM trial that alternated, with a roughly equal (±1) number of trials per bar phenotype. Model selection was then performed to acquire the best-fitted model from all possible combinations of variables according to the smallest corrected Akaike information criterion (AICc) using the "dredge" and "get. model" functions in the R package "MuMIn" [121]. In both top models, the variable "tank_side" was dropped and so excluded from the final (best) model. The quality of fitted GLMMs and misspecification was evaluated by assessing the simulated residual distributions using the R package DHARMa [105], which indicated no dispersion issues or major outliers.

### Ancestral state reconstruction

We performed stochastic character mapping [122] to re-analyze the possible evolutionary history of ontogenetic bar loss, as per Salis and colleagues [28]. To do so, we used the latest anemonefish species tree containing 28 species from Gaboriau and colleagues [30]. Stochastic mapping was performed using the function "make.simmap" in the R package Phytools [123]. First, we assessed the best model for the transition matrix from our empirical data. We fitted both a model with an equal rate of transition between states and another model with all variable rates using the function "ace" in the R package ape [124]. Comparisons of AIC scores between the two models supported the use of equal rates. The estimation

of ancestral state was also provided by the function "ace" in R package ape. For stochastic mapping, we computed 10,000 simulations using the best supported equal rates model and settings 'pi = "equal"' and 'Q = "empirical"'.

Note that our mapping of bar loss during ontogeny highlighted 10 species and three clades (Fig 6A), as opposed to the nine species and five clades previously reported in Salis and colleagues [28]. This discrepancy in clade number was due to the regrouping of *A. chrysopterus* and *A. latifasciatus* with *A. akindynos*/*A. mccullochi*, and *A. allardi*, respectively. In addition, *A. barberi* was previously miscategorized as not exhibiting ontogenetic bar loss [28], despite exhibiting two to three transient bars [125].

### Phylogenetic generalized least squares analyses

Because anemonefish color patterns are used in communicative signaling and influenced by the sea anemone symbiosis, we predicted that the presence of ontogenetic bar loss is associated with ecological factors, including group size (sociality) and host anemone species diversity. To test correlated evolutionary relationships between ontogenetic bar loss and (mean) group size, and host anemone species (total hosts, reproductive host), we performed PGLS regressions using a Brownian motion model. Due to the relatively recent (~10MY) adaptive radiation of anemonefishes, we also computed (non-phylogenetic) GLS regressions and compared AIC/BIC values. Both PGLS and GLS analyses were performed with the "gls" function from the R package nlme [126].

### Supporting information

**S1 Table. Ordbeta reg model output for white skin area in development experiment.** Letters denote environmental treatment, where "B"="Anemone"; "C"="Fake/plastic anemone"; and "D"="Empty".
(DOCX)

**S2 Table. Age-grouped comparisons of white skin area with adjusted coefficients for development experiment.** "*LCL*"=lower confidence limit; "*UCL*"=upper confidence limit. Letters denote environmental treatment, where "A"="Occupied Anemone"; "B"="Anemone"; "C"="Fake/plastic anemone"; and "D"="Empty".
(DOCX)

**S3 Table. Ordbeta reg model analyzing white skin loss in fish treated with Z-VAD-FMK.** Data excludes the outlier individual (C7).
(DOCX)

**S4 Table. GLMM output summary for Behavior Experiment 1.**
(DOCX)

**S5 Table. GLMM output summary for Behavior Experiment 2.**
(DOCX)

**S1 Fig. Juvenile body sizes in the developmental treatments.** Juvenile standard length per environmental treatment per replicate at 38 and 62 days-post-hatch/dph. Letters denote statistical significance (ANOVA, $p_{adj} < 0.05$) grouping. The data underlying this Figure can be found in https://doi.org/10.5281/zenodo.17973175.
(TIF)

**S2 Fig. Skin gene expression at 62 dph.** PCA showing the distribution of Day 44 (62 dph) body bar skin samples along PC1 and PC2 according to total gene expression, along with correlation matrix depicting pairwise Spearman's rank correlation coefficients, calculated using transcriptomic data. The data underlying this Figure can be found in https://doi.org/10.5281/zenodo.17973175.
(TIF)

**S3 Fig. TEM examination of pigmentation cells in adjacent orange skin patch.** Example of a TEM section in the orange skin, peripheral to the fading white bar, with a close-up showing pigment cells (triangles). No iridophores were detected. Bar plot shows the proportional abundance (counts) of different chromatophore types counted from 70 cells in three sections. The data underlying this Figure can be found in https://doi.org/10.5281/zenodo.17973175. (TIF)

**S4 Fig. Micrographs taken of TUNEL assay-treated skin sections taken in the fading white bar region.** Three Sections (10 μm thick) were made in dorsoventral regions of the fading body bar of *A. frenatus*, including brightfield "BF," TUNEL reaction mix-treated, and negative control. The side bar plot depicts the mean (±0.95 CI) area corrected total cellular fluorescence (CTCF) measured from individual cells across three sections per the ventral (section 1, $n = 64$), middle (section 2, $n = 75$), and dorsal (section 3, $n = 63$) regions. The data underlying this Figure can be found in https://doi.org/10.5281/zenodo.17973175. (TIF)

**S5 Fig. Micrographs of immunostained sections using cleaved/activated Caspase-3 antibody with DAB in the late stage, faded white skin.** No DAB+ cells were observed. The data underlying this Figure can be found in https://doi.org/10.5281/zenodo.17973175. (TIF)

**S6 Fig. PCA showing distribution of whole-body total transcriptomes for juvenile *A. frenatus* (n = 5), which cohabitated with or without an adult pair.** The data underlying this Figure can be found in https://doi.org/10.5281/zenodo.17973175. (TIF)

**S7 Fig. Individual counts of attacks made by (adult) male and female *A. frenatus* presented with juveniles.** Presented is the combined behavior data from Experiments 1 and 2. Boxes represent the median, 25th and 75th percentiles, and range (whiskers). Points represent the summed number of attacks per trial. The data underlying this Figure can be found in https://doi.org/10.5281/zenodo.17973175. (TIF)

**S8 Fig. Normalized transmission spectra measured through the clear plastic of the juvenile containers used in the behavior experiments.** Shown is the average transmission spectra ($n = 3$) with standard error shaded in gray. The data underlying this Figure can be found in https://doi.org/10.5281/zenodo.17973175. (TIF)

**S9 Fig. Landmark locations used in image analyses. (A)** Landmarks used for the alignment of fish images in color patch area quantification. **(B)** Vector locations for the mask used for isolating the body bar region were dorsally at the 8th dorsal spine and 4th dorsal ray, and in-line ventrally with the 2nd anal fin spine and posterior-edge of the pelvic fin. The data underlying this Figure can be found in https://doi.org/10.5281/zenodo.17973175. (TIF)

## Acknowledgments

We appreciate the service and support provided by members of the Okinawa Institute of Science and Technology (OIST) Sequencing Section for conducting transcriptome sequencing. We thank Marleen Klann for giving advice in TEM sample preparation. We also thank members from the OIST Scientific Imaging Section for providing support and training in use of the electron microscope and confocal microscope. We appreciate the OIST marine husbandry staff that assisted in the raising of fish, especially Danielle Miller and James Hutasoit. We give thanks to Kina Hayashi for input regarding

behavioral experiment design. We thank Marcela Herrera Sarrias and Stefano Vianello for advice on transcriptomic analyses. We thank Karen L. Cheney for lending spectrometry equipment. Our thanks to Florence Ruggiero, Jann Gibert, and Patrick Mehlen for their insightful conversations and advice. We also appreciate early manuscript feedback given by colleagues including Sam Reiter, Kai Sky Yu, Yi-Hsien Su, and Stephan Schneider. We thank Rachel C. Vella for kindly providing artwork.

## Author contributions

**Conceptualization:** Laurie J. Mitchell, Jann Zwahlen, Vincent Laudet.

**Data curation:** Laurie J. Mitchell, Vincent Laudet.

**Formal analysis:** Laurie J. Mitchell, Bruno Frédérich, Vincent Laudet.

**Funding acquisition:** Laurie J. Mitchell, Vincent Laudet.

**Investigation:** Laurie J. Mitchell, Saori Miura, Youjung Han, Jann Zwahlen, Camille A. Sautereau.

**Methodology:** Laurie J. Mitchell, Saori Miura, Jann Zwahlen, Camille A. Sautereau, Bruno Frédérich, Vincent Laudet.

**Project administration:** Laurie J. Mitchell.

**Resources:** Laurie J. Mitchell, Saori Miura, Camille A. Sautereau, Bruno Frédérich, Vincent Laudet.

**Validation:** Laurie J. Mitchell, Bruno Frédérich.

**Visualization:** Laurie J. Mitchell, Vincent Laudet.

**Writing – original draft:** Laurie J. Mitchell, Vincent Laudet.

**Writing – review & editing:** Laurie J. Mitchell, Saori Miura, Youjung Han, Jann Zwahlen, Camille A. Sautereau, Bruno Frédérich, Vincent Laudet.

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
