## [Editor Report · Decision Letter 0]

8 Aug 2025

Dear Dr Mitchell,

Thank you for submitting your manuscript entitled "Socially regulated developmental plasticity in the color pattern of an anemonefish" for consideration as a Research Article by PLOS Biology.

The PLOS Biology editorial staff have now had a chance to discuss your appeal, and we have decided to grant it based on input from an academic editor with relevant expertise. Therefore, we have reversed our initial decision and would now like to send your submission out for external peer review.

Once your full submission is complete, your paper will undergo a series of checks in preparation for peer review. After your manuscript has passed the checks it will be sent out for review. To provide the metadata for your submission, please Login to Editorial Manager (https://www.editorialmanager.com/pbiology) within two working days, i.e. by Aug 10 2025 11:59PM.

Kind regards,

Taylor

Taylor Hart, PhD,

Associate Editor

PLOS Biology

thart@plos.org

---

## [Decision Letter · Decision Letter 1]

22 Oct 2025

Dear Dr Mitchell,

Thank you for your patience while your manuscript "Socially regulated developmental plasticity in the color pattern of an anemonefish" was peer-reviewed at PLOS Biology. It has now been evaluated by the PLOS Biology editors and by several independent reviewers. We apologize again for the delay in reaching this decision.

In light of the reviews, which you will find at the end of this email, we would like to invite you to revise the work to thoroughly address the reviewers' reports.

As you will see, the reviewers wrote that the study is interesting, ambitious, and compelling. However, they also raised concerns about several aspects, including limitations in the behavioral data and the environmental treatment experiment, unacknowledged alternative explanations, insufficient rationale, and overstatements. In your revision, you should respond thoroughly to the reviewers' concerns.

Given the extent of revision needed, we cannot make a decision about publication until we have seen the revised manuscript and your response to the reviewers' comments. Your revised manuscript is likely to be sent for further evaluation by all or a subset of the reviewers.

**IMPORTANT - SUBMITTING YOUR REVISION**

*Re-submission Checklist*

*Published Peer Review*

*PLOS Data Policy*

*Blot and Gel Data Policy*

Sincerely,

Taylor

Taylor Hart, PhD,

Associate Editor

PLOS Biology

thart@plos.org

REVIEWS:

Reviewer #1: PBIOLOGY-D-25-02402R1

In this interesting, comprehensive, well-written and well-presented manuscript, Mitchell and colleagues use an integrative approach to investigate how social environment influences a key developmental trait: the loss of a white bar during the juvenile stage in an anemonefish. Juveniles living with adults lost their bar earlier than those kept in isolation, suggesting this is a socially-mediated form of phenotypic plasticity.

The researchers use a strong integrative approach to explore the mechanisms behind this change spanning transcriptomic analyses, transmission electron microscopy and pharmacological inhibition, that all pointed to the massive apoptosis of iridophores as a key mechanism in this color transition. Finally, he authors also explored the ecological and evolutionary implications of this plastic trait. Behavioral trials showed that adult anemonefish respond differently to juveniles with one versus two bars, but paradoxically adults showed heightened aggression to those that exhibited bar loss. They also used phylogenetic approaches to look at the evolution oof this traits and found that it is strongly associated with clownfish that form smaller social groups. The authors suggest that perhaps this developmentally plastic trait has some function during recruitment, perhaps mitigating conflict between recruit-stage juveniles and adults where there are fewer juveniles and thus less social buffering.

Overall, I was impressed by the extent and quality of the work, particularly the multifaceted work undertaken to explore mechanism. The molecular work is very solid and compelling - I really like the combination of transcriptomics, TM and pharmacological intervention to explore mechanism and I have no suggestions for improvement there.

However, there are a few things in the behavioral experimentation and conclusions that I was less convinced about or found unclear.

First, the authors do not make it particularly clear what the rationale is for the What is the rationale for the different environmental treatments. Presumably anemone and fake anenome are used to test for role of host on the color change, but I could not see this explicitly stated. I think it would be good to clarify this in revision.

Second, given the hypothesised importance of social environment to this color change process I was surprised that the "occupied" treatment was not replicated across the different settings. I think this is a significant weakness of that part of this study and it should be acknowledged by the authors in the results and discussion. To be clear, I am not advocating that the work be repeated with those additional treatments - that would be way too onerous, but the limitation should be dealt with explicitly in revision.

Realted to the above on line 205 the authors state "…our results suggest that the juvenile fish in response to perceiving their socially enriched environment accelerated their transition from the two-bar to the one-bar phenotype." This seems to be mostly supported by loss of bar stripes in the one social tank versus the others. Likely this finding is true, but replication of the occupied social state would strengthen. I note each tank setup was replicated n=2 times, but n=3 would likely have provided a stronger statistical framing.

I was puzzled by the experiment illustrated in figure 6, that shows, perhaps counter-intuitively, that juveniles that had transitioned to one bar were more likely to be attacked than the two barred individuals. The authors do not explore this as much as I thought they might, stating that this developmentally plastic trait might be linked to recruitment. I suspect this is probably on track, but is it not more likely that the two-bar morph is a sign of the juvenile stage and one-bar of the adult form? If it is, might you have observed more aggression in the occupied tank, and might the aggression, mediated via the stress axis, be a mechanism though which bar loss is accelerated? One way to tease this apart would be conduct experiments where individuals can see each other but not interact, to determine if it is aggressive interactions or just the visible presence of adult fish that accelerates the color change. I also wondered how you controlled for size and sex in these experiments?

Reviewer #2 [Sebastian Gaston Alvarado]: This is an ambitious and well-executed study that integrates molecular, cellular, behavioral, and phylogenetic approaches to demonstrate that ontogenetic bar loss in the tomato anemonefish is a socially regulated, plastic trait. The work convincingly establishes that the apoptosis of iridophores, mediated in part by thyroid hormone signaling, is the proximate mechanism for bar loss, while behavioral trials and evolutionary analyses highlight the ecological and evolutionary relevance of transient barring. The integration of these diverse datasets provides a compelling eco-evo-devo case study with broad significance for understanding plasticity in animal coloration.

That said, I have noted a handful of minor comments that can be addressed through clarifications in the text:

Line 586-596 (Comparative phylogenetics): "…revealed that species with bar loss during ontogeny have larger mean group sizes…"

Comment: The wording currently suggests causation. The authors should reframe this as a correlative association and acknowledge alternative ecological factors that could also influence group size and color pattern evolution.

Line 605(Discussion): "Here, we revealed environmentally induced temporal plasticity in a convergent, juvenile-to-adult reef fish color pattern transition."

Comment: This phrasing generalizes beyond the focal species. The authors should qualify that the mechanism was demonstrated specifically in Amphiprion frenatus, while suggesting that similar processes could occur across anemonefishes.

Line 610-611: "…increased TH production related gene expression and activity in the skin might form a neuroendocrinal link between the perception of social cues and shifts in chromatophore populations."

Comment: This implies direct hormone measurement, but the evidence is gene expression only. The authors should clarify that their inference of endocrine involvement is correlative, based on transcriptional signatures, rather than direct assays of thyroid hormone levels.

Overall, this study makes an important contribution by linking social context, cellular mechanisms, and evolutionary outcomes in a single system. By adding clarifying statements on sample sizes, species-specific scope, correlative framing, and the limits of gene-expression proxies for endocrine activity, the manuscript will present an appropriately cautious and balanced interpretation of the findings.

---

## [Editor Report · Decision Letter 2]

9 Jan 2026

Dear Dr Mitchell,

Thank you for your patience while we considered your revised manuscript "Socially regulated developmental plasticity in the color pattern of an anemonefish" for publication as a Research Article at PLOS Biology. This revised version of your manuscript has been evaluated by the PLOS Biology editors and the Academic Editor.

Based on our Academic Editor's assessment of your revision, we are likely to accept this manuscript for publication. Please also make sure to address the following data and other policy-related requests.

IMPORTANT: Please ensure that your next revision addresses the following editorial points:

--------------

**Title:

We suggest to modify the title to allude to the level of mechanistic insight in your study. Is the following alternative version acceptable to you?

"Iridophore apoptosis mediates socially-regulated developmental color pattern plasticity in an anemonefish"

**Financial disclosure statement:

-- Please add links to the funding agencies in the Financial Disclosure statement in the manuscript details.

**Ethics:

-- Please include the specific national or international regulations/guidelines to which your animal care and use protocol adhered. Please note that institutional or accreditation organization guidelines (such as AAALAC) do not meet this requirement.

**Supplement:

Please upload the supplementary figures separately, and please include the supplementary tables and their legends in the main manuscript file.

**Data:

-- Please upload the RNA sequencing data and make them available. Please update the statements in the submission form and in the paper to include the link.

-- Thank you for including the underlying data in your online supplement. We noticed that some of the items clearly state which figures they are associated with, (eg 'dev exp dataset.csv' is associated with Fig 1DE) but most of them do not. To increase accessibility of the supplemental items, we request that you add notes for the others. We appreciate that not all supplemental items may directly map onto figure plots, but please at least include notes for those that do. In particular, we require the numerical data underlying the following figure panels:

1D

3CDE

4BDE

5AB

6BCE

7CD

S1

S4

S7

-- Please also cite the location of the data clearly in all relevant main and supplementary Figure legends, e.g. “The data underlying this Figure can be found in https://doi.org/10.5281/zenodo.XXXXX”

--------------

We expect to receive your revised manuscript within two weeks.

*Published Peer Review History*

*Press*

Sincerely,

Taylor

Taylor Hart, PhD,

Associate Editor

thart@plos.org

PLOS Biology

---

## [Editor Report · Decision Letter 3]

16 Jan 2026

Dear Dr Mitchell,

Thank you for the submission of your revised Research Article "Iridophore apoptosis mediates socially-regulated developmental color pattern plasticity in an anemonefish" for publication in PLOS Biology. On behalf of my colleagues and the Academic Editor, Tom Baden, I am pleased to say that we can in principle accept your manuscript for publication, provided you address any remaining formatting and reporting issues. These will be detailed in an email you should receive within 2-3 business days from our colleagues in the journal operations team; no action is required from you until then. Please note that we will not be able to formally accept your manuscript and schedule it for publication until you have completed any requested changes.

PRESS

Sincerely,

Taylor

Taylor Hart, PhD,

Associate Editor

PLOS Biology

thart@plos.org